# BrainJanus: A Unified Model for Understanding and Generation across Brain, Vision, and Language

**Haitao Wu** [1 2 †]   **Qirui Zhang** [1]   **Zhouheng Yao** [2]   **Shangquan Sun** [2]   **Qihao Zheng** [2]   **Mianxin Liu** [2]   **Chi Zhang** [2]   **Wanli Ouyang** [2 3]   **Chunfeng Song** [2]   **Changqing Zhang** [1]   **Jiamin Wu** [2 3]

## Abstract

Modeling the bidirectional correspondence between external sensory stimuli and internal neural activity has emerged as a critical frontier in neuroscience. However, existing approaches predominantly treat brain encoding and decoding as isolated tasks, relying heavily on unimodal alignment and external priors while overlooking the brain's intrinsic nature as a multimodal integration system. To address these limitations, we propose BrainJanus, the first unified brain model that integrates brain, vision, and language within a single framework. Specifically, we introduce a Unified Brain Tokenizer to quantize continuous neural dynamics into discrete tokens aligned with visual and linguistic representations in a shared Omni space. Building on this, we utilize an All-in-One autoregressive architecture that leverages next-token prediction to enable seamless any-to-any generation, which encompasses image-to-brain and text-to-brain encoding, and brain-to-image and brain-to-text decoding. Extensive experiments demonstrate that BrainJanus achieves superior performance across diverse benchmarks. Furthermore, our framework exhibits zero-shot generalization and preserves interpretable biological topography, highlighting its potential as a general-purpose brain modeling paradigm. The code is available at GitHub.

† This work was done during his internship at Shanghai Artificial Intelligence Laboratory. [1]School of Artificial Intelligence, Tianjin University, Tianjin, China [2]Shanghai Artificial Intelligence Laboratory, Shanghai, China [3]The Chinese University of Hong Kong, Hong Kong, China. Correspondence to: Changqing Zhang <zhangchangqing@tju.edu.cn>, Jiamin Wu <wujiamin1@pjlab.org.cn>.

*Proceedings of the $43^{rd}$ International Conference on Machine Learning*, Seoul, South Korea. PMLR 306, 2026. Copyright 2026 by the author(s).

## 1. Introduction

Neuroscience once inspired the rise of deep learning, and now deep learning is in turn advancing our understanding of the brain, reflecting a cyclical inspiration between the two fields. In this context, modeling the relationship between external sensory stimuli and internal neural activity has become increasingly feasible (Horikawa & Kamitani, 2017; Shen et al., 2019; Takagi & Nishimoto, 2023; Allen et al., 2022), particularly for visual perception, and is essential for advancing brain-computer interfaces (Lorach et al., 2023).

Specifically, this relationship can be studied from two complementary directions. ***Brain encoding*** seeks to characterize how information from the external world is encoded in neural activity by learning models that predict brain responses from visual inputs (*image → brain*; Bao et al., 2025; Mai et al., 2025). Conversely, ***brain decoding*** aims to investigate how stimulus-related information can be inferred from neural activity, reconstructing perceived images (*brain → image*; Scotti et al., 2024; Takagi & Nishimoto, 2023) or generating textual descriptions (*brain → text*; Xia et al., 2024; Qiu et al., 2025) from neural signals.

Despite recent advancements, achieving high fidelity in these tasks necessitates a deeper understanding of neural representations. A critical gap exists between human perception mechanisms and current network paradigms. Neuroscientific studies (Huth et al., 2016; Ralph et al., 2017) indicate that the human brain is intrinsically a **system of multimodal integration**: semantic representations tile the entire cortex, extending from regions processing low-level visual features to higher-order conceptual domains. Thus, a visual stimulus elicits not only visual responses but also associated linguistic and semantic concepts (Binder et al., 2009). However, existing methods largely overlook this interplay. They predominantly rely on unimodal brain alignment with CLIP (Radford et al., 2021) visual features (Scotti et al., 2024; Lin et al., 2022; Bao et al., 2025) for both brain encoding and decoding, causing **insufficient exploitation** of multimodal semantics embedded in brain signals. Moreover, current methods typically treat brain visual encoding and decoding as isolated tasks with task-specific neural representations (Scotti et al., 2024; Wang et al., 2024b), despite

*Figure 1.* Illustration of the biological nature and the proposed modeling paradigm. (a) Biological Nature: The brain processes visual stimuli by projecting them into a unified multimodal space (Huth et al., 2016) that integrates both low-level pixel information and high-level semantic features. (b) Modeling Paradigm: Comparison between previous approaches and ours. Unlike previous task-specific, unidirectional pipelines that rely on separate aligners (e.g., CLIP) and generative models, our method employs a unified bidirectional autoregressive framework capable of performing both brain encoding and decoding within a single model.

the shared neural semantics inherent in bidirectional neural-stimuli correspondence. Consequently, this lossy modeling of neural signals necessitates an **over-reliance on large-scale external priors** (e.g., frozen pretrained CLIP, Latent Diffusion Models like Stable Diffusion (Rombach et al., 2022), or large language models such as LLaMA (Touvron et al., 2023) and GIT (Wang et al., 2022)) to compensate for the limited semantic content decoded from brain activity.

These observations motivate us to raise a critical question: **Can we build an omni model that unifies encoding and decoding across brain, vision, and language modalities?** As illustrated in Figure 1 (a), we posit that neural activity, vision, and language encode the same underlying semantics in different representational forms. To bridge these modalities, we propose **BrainJanus**, the first unified brain model capable of learning a unified token space across biological and digital domains. BrainJanus features a novel architecture comprising: (1) **Unified Brain Tokenizer**: We design a brain tokenizer trained from scratch to quantize continuous neural dynamics into discrete tokens. By aligning these with vision and language tokens, we map all modality signals into a shared Omni space; and (2) **All-in-One Autoregressive Model**: We utilize a single Transformer backbone that generalizes cross-modal interactions via next-token prediction. This enables the model to seamlessly toggle between four distinct tasks, encoding images and text into neural representations and decoding neural signals back into visual and textual formats, all within a unified framework.

Our main contributions are summarized as follows:

- We propose BrainJanus, the first unified autoregressive framework that bridges brain, vision, and language via a shared discrete token space, enabling seamless

any-to-any generation within a single model.

- BrainJanus achieves competitive performance on a wide range of encoding and decoding benchmarks, and consistently outperforms task-specific models under joint training.

- Unified multi-task learning promotes effective cross-modal knowledge transfer, allowing BrainJanus to perform zero-shot generalization with strong task-agnostic representations.

- We further show that the generated fMRI signals preserve interpretable cortical topography and biological variability, indicating that BrainJanus captures meaningful neural representations.

## 2. Related Work

### 2.1. Neural Representation Pretraining

Recent advances in neural representation pretraining have leveraged large-scale data and transformer architectures to explore diverse neuroimaging modalities. In the domain of EEG, models such as LaBraM (Jiang et al., 2024), EEGPT (Wang et al., 2024a), and EEGformer (Chen et al., 2024) employ self-supervised or generative frameworks to enhance temporal modeling capacity and cross-subject generalization. Parallel efforts in fMRI, including BrainLM (Caro et al., 2023) and MindEye2 (Scotti et al., 2024), focus on bridging neural activity with semantic and visual latent spaces for reconstruction and alignment tasks. More recently, unified frameworks like BrainOmni (Xiao et al., 2025) and BrainFLORA (Li et al., 2025b) have attempted to integrate heterogeneous signals (e.g., EEG,

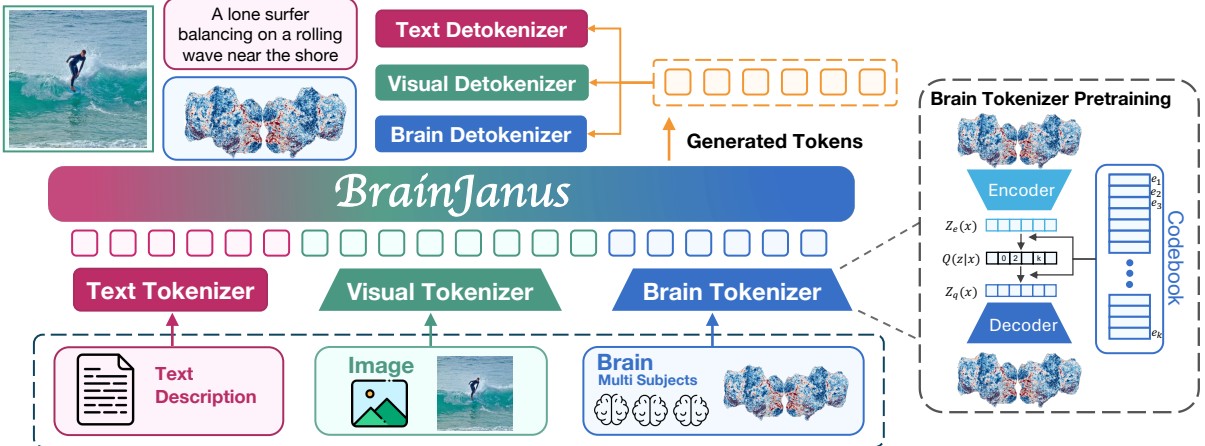

*Figure 2.* An overview of BrainJanus. The input data, regardless of its modality, is tokenized into a shared token space and then organized into a token sequence. BrainJanus processes these tokens autoregressively, enabling arbitrary transformations among brain, vision, and language modalities.

*Table 1.* Comparison of brain decoding and encoding capabilities across existing methods.

| Method | Brain Decoding | | Brain Encoding | |
|---|---|---|---|---|
| | To Img | To Text | From Img | From Text |
| Mind Reader (Lin et al., 2022) | ✓ | ✓ | ✓ | ✗ |
| BrainDiffuser (Ozcelik & VanRullen, 2023) | ✓ | ✗ | ✗ | ✗ |
| SDRecon (Takagi & Nishimoto, 2023) | ✓ | ✓ | ✗ | ✗ |
| BrainCap (Ferrante et al., 2023) | ✓ | ✓ | ✗ | ✗ |
| MinD-Vis (Chen et al., 2023) | ✓ | ✗ | ✗ | ✗ |
| UniBrain (Mai & Zhang, 2023) | ✓ | ✓ | ✗ | ✗ |
| MindEye2 (Scotti et al., 2024) | ✓ | ✓ | ✗ | ✗ |
| OneLLM (Han et al., 2024) | ✗ | ✓ | ✗ | ✗ |
| UMBRAE (Xia et al., 2024) | ✓ | ✓ | ✗ | ✗ |
| MindBridge (Wang et al., 2024b) | ✓ | ✗ | ✗ | ✗ |
| MindSimulator (Bao et al., 2025) | ✗ | ✗ | ✓ | ✗ |
| SynBrain (Mai et al., 2025) | ✗ | ✗ | ✓ | ✗ |
| MindLLM (Qiu et al., 2025) | ✗ | ✓ | ✗ | ✗ |
| BrainFLORA (Li et al., 2025b) | ✓ | ✓ | ✗ | ✗ |
| **BrainJanus (Ours)** | ✓ | ✓ | ✓ | ✓ |

MEG, and fMRI) into shared representations. Nevertheless, prior work has not yet developed a universal brain tokenizer capable of converting continuous neural recordings into discrete tokens and unifying these neural tokens with text and vision tokens within a shared token space for joint modeling.

## 2.2. Neural Encoding and Decoding

Recent advances have significantly advanced bidirectional mapping between neural activities and perceptual stimuli, despite the low signal-to-noise ratio of brain signals. In visual decoding, research emphasizes high-fidelity reconstruction and semantic alignment. To address cross-subject variability, MindEye2 (Scotti et al., 2024) employs shared-subject modeling, while CLIP-MUSED (Zhou et al., 2024b) uses CLIP-guided training. UniBrain (Mai & Zhang, 2023) integrates reconstruction and captioning in a latent diffusion framework for improved generative quality. BraVL (Du

et al., 2023) and NICE (Song et al., 2023) apply contrastive learning to align fMRI and EEG signals with visual-linguistic features, enabling zero-shot recognition. Recent works also incorporate Multimodal Large Language Models: UMBRAE (Xia et al., 2024) and MindLLM (Qiu et al., 2025) align brain encoders with LLMs to support diverse tasks, from grounding to open-ended instruction tuning. In neural encoding, MEG-GPT (Huang et al., 2025) provides a transformer-based foundation model for MEG, while SynBrain (Mai et al., 2025) uses probabilistic learning to synthesize fMRI signals, improving decoding in data-scarce settings. Nevertheless, most studies treat decoding and encoding as independent processes, and no unified framework yet achieves simultaneous bidirectional mapping.

## 2.3. Unified Understanding and Generation Model

Unified multimodal understanding and generation models aim to integrate perception and synthesis across text, image, video, and audio within a single architecture. Recent advancements have explored various strategies to achieve this unification. Models like Chameleon (Chameleon Team, 2024) and Emu3 (Wang et al., 2024c) validate that discrete tokenization allows different modalities to be modeled jointly through pure autoregressive next-token prediction. To further enhance synthesis quality alongside understanding, Show-o (Xie et al., 2024) and Transfusion (Zhou et al., 2024a) integrate discrete or continuous diffusion mechanisms directly into the transformer training objectives. Furthermore, works such as NExT-GPT (Wu et al., 2024), Janus-Pro (Chen et al., 2025), and BAGEL (Deng et al., 2025) focus on flexible any-to-any frameworks and large-scale pretraining to unlock emergent reasoning and instruction-following capabilities. However, these unified paradigms are currently confined to external sensory data. Extending this unification to bridge internal brain signals with vision

and language remains an unexplored frontier.

# 3. Method

In this section, we first introduce the basic notation and then describe the overall framework of BrainJanus. The proposed method consists of two main stages. In the first stage, we perform unified brain tokenizer pretraining, which maps brain signals into a shared Omni space. In the second stage, we perform supervised fine-tuning (SFT) on a Transformer backbone, jointly training it on four tasks that involve encoding and decoding across brain signals, vision, and language in a mixed manner.

## 3.1. Preliminaries

We consider three modalities of raw data $(\mathcal{B}, \mathcal{V}, \mathcal{L})$, where $\mathcal{B}$, $\mathcal{V}$, and $\mathcal{L}$ denote the spaces of brain signals (e.g., fMRI or EEG recordings), visual inputs (e.g., natural images), and linguistic inputs (e.g., text sequences), respectively. For a given sample, we denote the tri-tuple $(x_B, x_V, x_L) \in \mathcal{B} \times \mathcal{V} \times \mathcal{L}$, where $x_B$, $x_V$, and $x_L$ correspond to the raw brain, vision, and language data.

## 3.2. Unified Brain Tokenizer Pretraining

In order to map continuous neural dynamics into a discrete format compatible with other modalities, we train a Unified Brain Tokenizer from scratch to convert continuous neural signals into a sequence of discrete tokens that can be seamlessly integrated into the unified Omni space. Specifically, we introduce a discrete codebook $\mathcal{C} = \{e_k\}_{k=1}^K$, where each code vector $e_k \in \mathbb{R}^{d_\mathcal{O}}$ represents a token embedding. Given a neural input $x$, the encoder produces a continuous latent representation $z_e(x)$, which is then quantized by nearest-neighbor lookup in the codebook:

$$z_q(x) = \arg \min_{c \in \mathcal{C}} \|z_e(x) - c\|_2^2. \tag{1}$$

We adopt a VQ-style autoencoding objective and optimize the following loss:

$$\mathcal{L}_T = \log p(x \mid z_q(x)) + \left\| \text{sg}[z_e(x)] - e_k \right\|_2^2 + \beta \left\| z_e(x) - \text{sg}[e_k] \right\|_2^2, \tag{2}$$

where $\text{sg}(\cdot)$ denotes the stop-gradient operator. The loss function consists of three terms. The reconstruction term $\log p(x \mid z_q(x))$ encourages the decoder to accurately reconstruct the input $x$ from the quantized latent $z_q(x)$. The codebook term moves the selected codebook vector $e_k$ toward the encoder output, thereby updating the codebook. The commitment term penalizes the encoder for large deviations from its assigned codebook embedding, improving training stability.

**Tokenization of other modalities.** To unify multimodal inputs under a single autoregressive modeling framework, we additionally leverage established tokenizers for vision and text. For the visual modality, we use the image tokenizer from Sun et al. (2024) to map an input image into a sequence of discrete IDs. For the text modality, we use the tokenizer from Chen et al. (2025) to obtain subword-level token IDs. For each modality, the resulting ID sequence is flattened into a 1-D token stream, enabling arbitrary interleaving across modalities during training and inference.

## 3.3. Unified Token Generation

To mitigate the heterogeneity across different modalities, including brain, vision, and language, we project all raw inputs into a shared representation space, referred to as the *Omni space* $\mathcal{O}$. Formally, $\mathcal{O}$ is defined as the space of finite-length token sequences, where each token lies in a fixed-dimensional embedding space of dimension $d_\mathcal{O}$:

$$\mathcal{O} := \bigcup_{n \geq 1} \left( \mathbb{R}^{d_\mathcal{O}} \right)^n, \tag{3}$$

where $n$ denotes the number of tokens in the sequence. For each modality $\phi \in \{B, V, L\}$, we introduce a modality-specific tokenizer (or encoder) $f_\phi : \mathcal{X}_\phi \to \mathcal{O}$ that maps raw inputs from its corresponding domain into the Omni space:

$$f_B : \mathcal{B} \to \mathcal{O}, \quad f_V : \mathcal{V} \to \mathcal{O}, \quad f_L : \mathcal{L} \to \mathcal{O}. \tag{4}$$

Given a multimodal input triplet $(x_B, x_V, x_L)$, each modality is independently encoded into a sequence of token embeddings with a shared dimensionality but potentially different sequence lengths:

$$\begin{aligned} b &= f_B(x_B) = (b_1, \ldots, b_{n_B}), \\ v &= f_V(x_V) = (v_1, \ldots, v_{n_V}), \\ l &= f_L(x_L) = (l_1, \ldots, l_{n_L}), \end{aligned} \tag{5}$$

where $b, v, l \in \mathcal{O}$, and $n_B$, $n_V$, and $n_L$ denote the sequence lengths, which may vary across samples. By aligning heterogeneous modalities into the same token space, this formulation enables a single unified model to process multimodal inputs in a seamless and architecture-agnostic manner.

**Unified Autoregressive Modeling.** On top of the Omni space, we define a unified autoregressive model $p_\theta$ that operates over interleaved multimodal token sequences. Given a prefix of tokens $(z_1, \ldots, z_{t-1}) \in \mathcal{O}$, the model predicts the next token according to $p_\theta(z_t \mid z_{<t})$, where $z_{<t} \triangleq (z_1, \ldots, z_{t-1})$ denotes the multimodal context. Notably, each context token may originate from any modality, i.e., for each $i < t$, $z_i \in \mathcal{O}$, and tokens from different modalities can be arbitrarily interleaved within the sequence. The model is trained by minimizing the negative log-likelihood

*Table 2.* Quantitative results of brain-to-caption generation on two ground-truth settings (COCO captions and detailed Qwen-generated captions). Using Qwen captions as GT may underestimate prior methods trained on original COCO captions, but yields better image-text semantic alignment (see Figures 7 and 9). **Bold** values indicate the best performance.

| Method | BLEU1↑ | BLEU2↑ | BLEU3↑ | BLEU4↑ | METEOR↑ | ROUGE↑ | CIDEr↑ | SPICE↑ | BERT↑ | CLIP↑ |
|---|---|---|---|---|---|---|---|---|---|---|
| **GT = COCO Caption (Default, Short)** | | | | | | | | | | |
| MindEye2 | 57.51 | 38.55 | 25.63 | 18.21 | 18.53 | 40.21 | 58.21 | 9.21 | 40.12 | 91.7% |
| UMBRAE | 59.44 | 40.48 | 27.66 | 19.03 | 19.45 | 43.71 | 61.06 | 12.79 | 40.67 | 92.5% |
| MindLLM | 61.75 | 42.84 | 29.86 | 21.24 | 19.54 | 45.82 | 60.97 | 11.79 | 40.42 | 92.1% |
| **BrainJanus** | **63.20** | **44.10** | **31.25** | **22.45** | **20.21** | **46.91** | **62.37** | **13.10** | **42.12** | **94.8%** |
| **GT = Qwen Caption (Detailed)** | | | | | | | | | | |
| MindEye2 | 4.32 | 1.81 | 0.82 | 0.38 | 5.41 | 18.12 | 1.03 | 8.21 | 29.12 | 93.0% |
| UMBRAE | 8.31 | 3.62 | 1.67 | 0.77 | 6.37 | 19.15 | 2.46 | 9.05 | 30.91 | 94.7% |
| MindLLM | 9.53 | 4.27 | 1.83 | 0.91 | 6.01 | 20.33 | 2.11 | 6.63 | 30.62 | 88.9% |
| **BrainJanus** | **40.21** | **21.12** | **11.63** | **6.74** | **14.01** | **29.43** | **41.32** | **11.71** | **38.12** | **96.2%** |

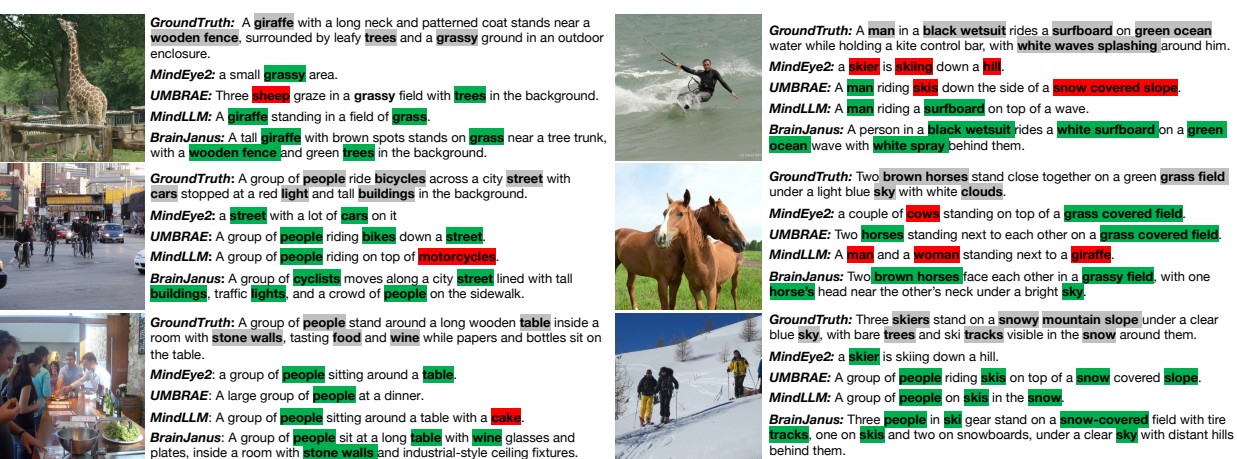

*Figure 3.* Qualitative comparison of brain caption decoding results. GroundTruth image captions are compared with captions decoded from fMRI voxel signals using MindEye2, UMBRAE, MindLLM, and BrainJanus (ours). Gray indicates key objects, Green highlights indicate semantic matches with the GroundTruth, while red highlights denote errors. More results are shown in Figure 9.

over the full multimodal token sequence $(z_1, \ldots, z_T)$:

$$\mathcal{L}(\theta) = -\sum_{t=1}^{T} \log p_\theta(z_t \mid z_{<t}). \quad (6)$$

By representing all modalities in a common Omni space, the model enables seamless bidirectional generation across arbitrary modality pairs, including brain ↔ image, brain ↔ text, and image ↔ text. Concretely, the same model can perform translation, completion, and conditional generation by treating any modality tokens as context and autoregressively sampling remaining tokens, allowing flexible interleaving and multi-step cross-modal composition (e.g., brain → text → image) without training separate models.

## 4. Experiments

### 4.1. Experimental Setup

**Datasets.** We conduct experiments on the Natural Scenes Dataset (NSD) (Allen et al., 2022), a large-scale fMRI

dataset where 8 subjects viewed natural images from the COCO dataset (Lin et al., 2014) across approximately 40 hours of scanning. In line with MindEye2 (Scotti et al., 2024), we focus on 4 subjects (Sub-1, Sub-2, Sub-5, Sub-7) for evaluation who completed all experimental sessions. For each subject, we use 9,000 unique images for training and evaluate the model on a shared set of 1,000 test images, each presented across 3 trials to account for response variability. Prior studies (Lin et al., 2022; Ferrante et al., 2023; Mai & Zhang, 2023; Scotti et al., 2024) have primarily relied on original COCO captions as the ground truth. However, as shown in Figure 7 and Table 3, quantitative and qualitative analysis indicates that these annotations often suffer from limitations such as brevity and lack of detail. To address this, we leveraged several state-of-the-art multimodal large language models such as Qwen3-VL-235B-A22B-Instruct (Yang et al., 2025) to synthesize highly detailed and comprehensive captions for 73k images. By measuring semantic similarity, we show that the generated captions better match the images than the original annotations (Figure 7).

*Table 3.* Quantitative comparison of image reconstruction quality. We compare BrainJanus, the only autoregressive model for direct brain-to-image generation, with prior diffusion-based SOTA methods. **Bold** and underlined indicate the best and second-best results. **Zero-shot** denotes training only on brain-to-text pairs.

| Method | PixCorr↑ | SSIM↑ | Alex(2)↑ | Alex(5)↑ | Incep↑ | CLIP↑ | Eff↓ | SwAV↓ |
|---|---|---|---|---|---|---|---|---|
| **Caption to Image** | | | | | | | | |
| Coco Caption | 0.097 | 0.28 | 77.7% | 90.0% | 95.2% | 96.3% | 0.656 | 0.437 |
| GIT-large | 0.091 | 0.285 | 77.8% | 89.6% | 95.5% | 95.8% | 0.681 | 0.446 |
| Qwen3-VL-235B-IT | 0.153 | 0.285 | 85.0% | 94.2% | 95.4% | 96.7% | 0.607 | 0.402 |
| **CLIP + unCLIP + Diffusion Prior + GIT Model** | | | | | | | | |
| Takagi and Nishimoto | - | - | 78.9% | 85.6% | 83.8% | 82.1% | 0.811 | 0.504 |
| Ozcelik and VanRullen | 0.273 | 0.365 | 94.4% | 96.6% | 91.3% | 90.9% | 0.728 | 0.421 |
| UMBRAE | 0.283 | 0.341 | 95.5% | 97.0% | 91.7% | 93.5% | 0.700 | 0.393 |
| MindEye2 | **0.322** | **0.431** | **96.1%** | **98.6%** | **95.4%** | 93.0% | **0.619** | **0.344** |
| **Autoregressive Model** | | | | | | | | |
| BrainJanus (Zero-shot) | 0.077 | 0.257 | 69.3% | 77.1% | 77.5% | 77.3% | 0.891 | 0.564 |
| BrainJanus | 0.173 | 0.292 | 90.3% | 97.1% | 94.7% | **94.4%** | 0.656 | 0.372 |

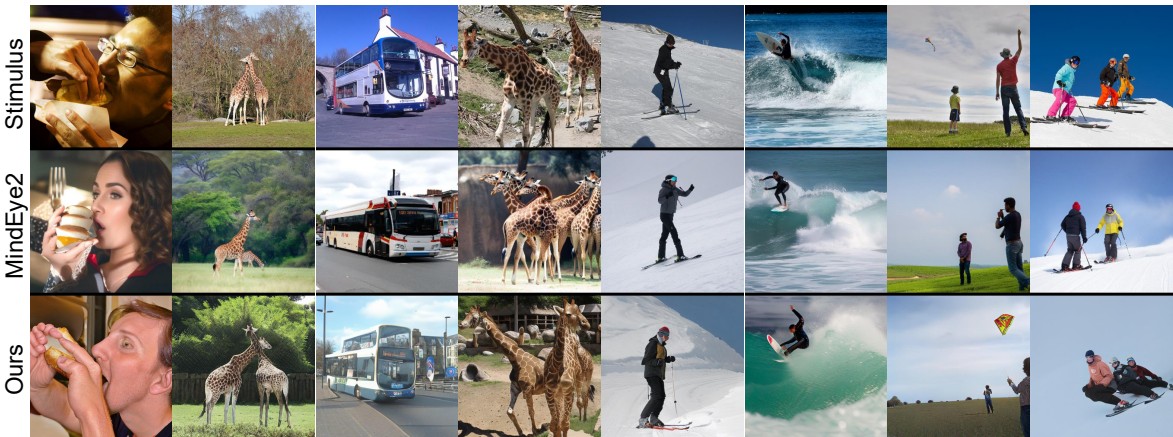

*Figure 4.* Qualitative comparison of visual decoding for Subject 1. Our method outperforms MindEye2 by generating reconstructions with higher semantic accuracy, better preservation of object and action attributes, and improved structural consistency across diverse visual categories. More examples can be found at Figure 10.

**Implementation Details.** For the brain tokenizer, we adopt a VQ-VAE architecture with a codebook size of 128, a compression ratio of 128, and an embedding dimension of 32. The tokenizer is trained using the AdamW (Loshchilov & Hutter, 2017) optimizer with an initial learning rate of $1 \times 10^{-4}$, a batch size of 256, and for 100 epochs. We set the commitment loss coefficient to 0.25 and the entropy loss ratio to 0.1. The training is performed in a cross-subject setting with 8 subjects. For autoregressive modeling, we employ the AdamW optimizer with a cosine learning rate schedule and warmup, where the peak learning rate is set to $2 \times 10^{-4}$. The model is trained for 15 epochs with a batch size of 16, using ZeRO Stage-2 optimization. The backbone is initialized with the parameters of Janus-7B, and the hidden dimension is set to 4096. We adopt LoRA (Hu et al., 2022) for parameter-efficient fine-tuning, applying low-rank adapters to the query and value projections with a scaling factor of 16 and a dropout rate of 0.2, while keeping all other parameters frozen. During this stage, other components are frozen.

More details on data preprocessing, hyperparameter settings, software versions and hardware configurations are provided in Section A.

**Metrics for Brain to Image Decoding.** Consistent with established protocols in brain visual reconstruction (Scotti et al., 2024; Wang et al., 2024b), we assess the quality of generated images using a comprehensive suite of metrics categorized into low-level structural fidelity and high-level semantic alignment. For low-level evaluation, we report Pixel Correlation (PixCorr) and Structural Similarity Index Measure (SSIM) (Wang et al., 2004) to quantify pixel-wise accuracy and structural consistency. Additionally, we utilize the early layers of AlexNet (Layer 2 and Layer 5) (Krizhevsky et al., 2012) to evaluate the reconstruction of basic visual features such as edges and textures. For high-level semantic evaluation, we employ feature similarity metrics based on deep neural networks, including

| Stimulus | Raw fMRI | Syn fMRI | Syn to Image |
|---|---|---|---|

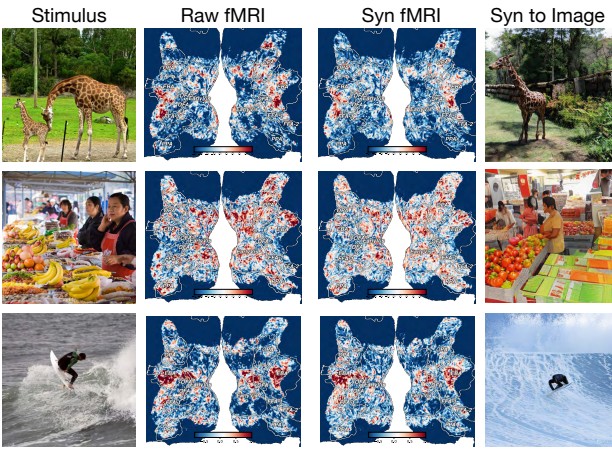

*Figure 5.* Qualitative result of brain encoding. Additional examples are provided in the appendix (see Figure 11).

Inception(Szegedy et al., 2016), EfficientNet (EffNet) (Tan & Le, 2019), and SwAV (Caron et al., 2020). Furthermore, we rely on CLIP (Radford et al., 2021) to measure the semantic correspondence between the reconstructed images and the ground truth, ensuring the conceptual content is accurately preserved.

**Metrics for Brain to Text Decoding.** Following previous works in brain captioning (Li et al., 2025a; Xia et al., 2024; Qiu et al., 2025; Li et al., 2025b), we evaluate our method with standard metrics including BLEU-$k$ (Papineni et al., 2002), ROUGE (Lin, 2004), CIDEr (Vedantam et al., 2015), SPICE (Anderson et al., 2016), METEOR (Banerjee & Lavie, 2005). Additionally, we report BERTScore (Zhang et al., 2019), CLIP-Text and CLIP-Image score (Radford et al., 2021).

**Metrics for Brain Encoding.** Although predicting neural activity from visual stimuli remains highly challenging, recent methods report strong performance on voxel-level and semantic-level metrics that may sometimes reflect information leakage or superficial statistical regularities rather than faithful neural modeling. We discuss these limitations and their implications in detail in Section 4.3.

## 4.2. Brain Decoding

We conduct brain decoding experiments to evaluate the ability of BrainJanus to decode both textual captions and visual images from brain signals. Quantitative and qualitative comparisons with baselines are presented below. Additional results are provided in Section B.2.

**Quantitative Comparison.** BrainJanus substantially outperforms prior methods on brain-to-text decoding, achieving a BERTScore of 38.12 and a CLIP score of 96.2%, surpassing the previous state-of-the-art by 7.21 and 1.5%, respectively (see Table 2). On brain-to-image decoding, BrainJanus, which is the only autoregressive approach

among the compared methods, achieves the highest CLIP semantic similarity of 94.4% despite using diffusion-free generation, outperforming diffusion-based baselines in high-level alignment(see Table 3). Notably, BrainJanus also exceeds pure caption-to-image baselines on low-level fidelity metrics, indicating that brain signals preserve richer low-level visual information than textual captions alone.

**Qualitative Comparison.** As shown in Figure 3 and Figure 4, BrainJanus generates more accurate, detailed, and semantically faithful textual descriptions than prior methods. For visual reconstruction, our autoregressive outputs exhibit stronger preservation of object attributes, actions, and overall scene structure compared to diffusion-based baselines (see Figure 4).

## 4.3. Brain Encoding

Before addressing this challenging task, we must first establish how to evaluate the quality of synthetic fMRI. While voxel-level (Gu et al., 2022; Wang et al., 2023) and semantic-level (Bao et al., 2025; Mai et al., 2025) metrics are the standard protocols, they both face significant issues. Additional results are provided in Section B.3.

**Voxel-level variability.** Previous methods rely on voxel-wise metrics, such as Pearson correlation and Mean Squared Error (MSE). However, these local measures suffer from two main drawbacks. First, they ignore global structure, often failing to penalize predictions that lack spatial coherence. Second, they are overly sensitive to the natural variability of neural responses across trials. To analyze this, we examined the distribution of MSE, Pearson correlation, and cosine similarity across three trials of the same stimuli for eight subjects (Figures 12 to 14). Importantly, these inter-trial consistency metrics establish a noise ceiling, which represents the maximum possible performance for any encoding model. Our statistical analysis shows that biological variability places strict limits on performance: the empirical noise floor for MSE is approximately **0.55**, while the upper bounds for Cosine Similarity and Pearson Correlation remain below **0.65**.

**Semantic-level Hacking.** Semantic-level evaluation aims to bypass low-level noise by measuring the quality of images reconstructed from synthesized brain signals. This typically involves an encoding-decoding pipeline (Image → Visual Embedding → Syn-fMRI → Reconstructed Image). However, we show that this protocol is vulnerable to a trivial exploit. We introduce a simple baseline, the Padding Hacking (illustrated in Figure 16). Instead of learning a real biological mapping, this method simply zero-pads the ground truth visual embedding to match the voxel size $N$, effectively treating the brain region as a sparse storage container. The decoder then reverses this to retrieve the exact embedding. As shown in Table 4, this biologically mean-

*Table 4.* Evaluation hacking analysis of visual-to-fMRI synthesis. Metrics are grouped into *Low-Level* (structural/perceptual) and *High-Level* (semantic). A trivial Padding Hacking baseline achieves near-perfect scores by leaking visual embeddings, showing that image reconstruction metrics alone are insufficient to validate neural synthesis quality.

| Method | Low-Level (Pixel & Structure) | | | | High-Level (Semantic) | | | |
| --- | --- | --- | --- | --- | --- | --- | --- | --- |
| | PixCorr ↑ | SSIM ↑ | Alex(2) ↑ | Alex(5) ↑ | Incep ↑ | CLIP ↑ | Eff ↓ | SwAV ↓ |
| MindSimulator (Bao et al., 2025) | 0.201 | 0.298 | 89.6% | 96.8% | 93.2% | 91.2% | 0.688 | 0.393 |
| SynBrain (Mai et al., 2025) | - | - | - | - | 95.7% | 94.3% | 0.639 | 0.362 |
| **Padding Hacking (Ours)** | **0.919** | **0.595** | **100.0%** | **100.0%** | **100.0%** | **100.0%** | **0.104** | **0.045** |

*Table 5.* Brain encoding performance comparison. We train with only MSE or CE loss to avoid hacking.

| | Image | | Text | |
| --- | --- | --- | --- | --- |
| Voxel Source | Inception ↑ | CLIP ↑ | CLIP ↑ | BERT ↑ |
| GT Voxel (upper bound) | 94.7% | 94.4% | 96.2% | 38.1 |
| Linear Regressive | 65.9% | 68.5% | 72.5% | 25.1 |
| Transformer Encoding | 70.4% | 72.4% | 73.9% | 26.4 |
| **BrainJanus** | **72.8%** | **75.3%** | **77.0%** | **28.3** |

ingless approach achieves perfect scores across all metrics. This result reveals a critical flaw: the evaluation cannot distinguish between true neural synthesis and simple information leakage. Therefore, to ensure validity, training methods must avoid direct alignment with pre-trained visual embeddings.

**Quantitative Comparison.** In contrast to prior methods that may be susceptible to information leakage in semantic-level evaluation, our approach relies solely on cross-entropy loss for the encoding task. This design encourages the model to learn meaningful neural patterns rather than exploiting superficial feature storage. As shown in Table 5, BrainJanus outperforms various baselines across semantic metrics, demonstrating that our synthesized fMRI signals preserve meaningful semantic information through genuine learning.

**Qualitative Comparison.** Qualitative results, shown in Figure 5, illustrate that images reconstructed from our synthesized fMRI signals retain recognizable semantic content and visual characteristics, further confirming the effectiveness of our bidirectional mapping.

### 4.4. Ablation Study

**Trade-off between Compression Ratio and Semantic Preservation.** We empirically observe a clear trade-off between compression efficiency and semantic preservation in discrete representation learning. Specifically, lower compression ratios and larger codebooks retain more information, leading to better reconstruction quality and semantic consistency. However, these benefits result in significantly longer token sequences, which increases the difficulty of autoregressive generation. To analyze this balance, we conduct ablation studies on compression ratios and codebook

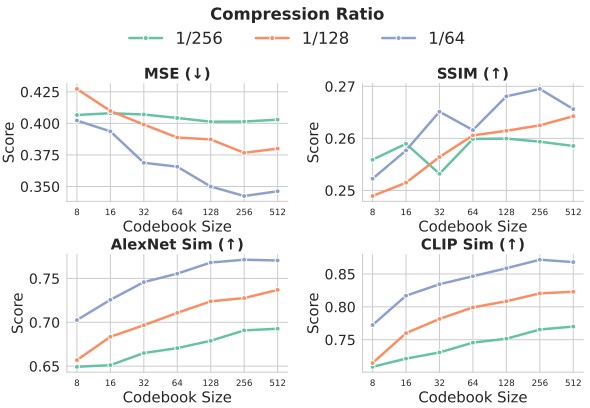

*Figure 6.* Ablation results of the brain tokenizer under different codebook sizes and compression ratios. We report reconstruction fidelity (MSE, SSIM), intermediate feature similarity (AlexNet2), and high-level semantic alignment (CLIP). The results reveal a clear trade-off between compression and information preservation.

sizes. We evaluate the representations in terms of codebook usage, reconstruction quality, alignment, and downstream performance (Figure 6), with qualitative examples shown in Figure 15. Results indicate that increasing the codebook size beyond 128 or 256 offers only limited gains. Notably, the compression process not only preserves semantics but also acts as a filter, effectively removing high-frequency noise. **Zero-shot Cross-task Generalization.** For the decoding tasks, a model trained exclusively on fMRI-to-text generation can be applied to fMRI-to-image generation in a zero-shot manner, and the same observation holds in the reverse direction. These results suggest that the learned representations generalize across decoding tasks. Quantitative results are reported in Table 3.

## 5. Conclusion

In this work, we propose BrainJanus, the first unified brain model that integrates brain encoding and decoding with vision and language via a shared discrete token space and a single autoregressive Transformer, enabling seamless any-to-any generation across modalities. BrainJanus achieves superior performance on diverse encoding and decoding benchmarks, outperforms task-specific models under joint training, and demonstrates strong zero-shot generalization.

It also generates biologically plausible fMRI signals that preserve interpretable cortical topography and individual variability.

**Limitation.** BrainJanus is currently restricted to fMRI data within the visual cortex, rather than whole-brain activity. Additionally, the reliance on powerful generative priors may introduce "hallucinations," where the model prioritizes visual quality over strict biological faithfulness. Finally, the high computational cost and its robustness across more diverse neural modalities and subject populations remain to be fully explored.

**Acknowledgements.** This work is partially supported by the National Natural Science Foundation of China (Grant No.62376193, Grant No.62573414). This work was supported by Shanghai Artificial Intelligence Laboratory and the JC STEM Lab of AI for Science and Engineering, funded by The Hong Kong Jockey Club Charities Trust, the Research Grants Council of Hong Kong (Project No. CUHK14213224). This work is supported by Shanghai Artificial Intelligence Laboratory, and the Lingang Laboratory (Grant No. LGL-1987-19). The authors appreciate the valuable feedback from anonymous reviewers.

## Impact Statement

This paper presents work whose goal is to advance the field of Multimodal Learning across brain, vision, language. There are many potential societal consequences of our work, none which we feel must be specifically highlighted here.

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

# A. Experimental details

## A.1. Natural Scenes Dataset

We conduct experiments on the *Natural Scenes Dataset* (NSD), the largest publicly available dataset pairing high-resolution fMRI recordings with natural image stimuli. NSD provides extensive 7T fMRI data collected from eight subjects viewing images sampled from the COCO dataset. In each trial, an image was presented for 3 seconds, followed by a task in which the subject reported whether the image had been previously seen during the experiment. Following prior work, we use data from 8 subjects for training and 4 subjects for final testing. To ensure data completeness and consistency, we restrict our analysis to four subjects (subj01, subj02, subj05, and subj07) who completed all image-viewing sessions. For each subject, the dataset is divided into a training set comprising 9,000 unique images (27,000 fMRI trials) and a test set comprising 1,000 images presented three times each (3,000 fMRI trials). Notably, while training images differ across subjects, the test images are shared across all subjects, enabling evaluation on a common stimulus set.

We use the preprocessed fMRI data released with NSD, at a spatial resolution of 1.8 mm. Neural responses are represented as single-trial beta weights estimated via generalized linear models (GLMs), capturing trial-specific activation patterns. All analyses are performed within predefined regions of interest (ROIs) provided by NSD, covering early visual cortex and higher-level ventral visual areas. The number of voxels in the selected ROIs for subj01, subj02, subj05, and subj07 is 15,724, 14,278, 13,039, and 12,682, respectively. Further details on fMRI acquisition, preprocessing, and ROI definitions can be found in (Allen et al., 2022). Previously, COCO Captions were commonly used as ground truth, but they are often shorter and lack details. We generated more detailed descriptions based on the latest multimodal large language models Figure 9, and computed the similarities between three types of captions (COCO (Lin et al., 2014), GIT (Wang et al., 2022)), and Qwen3-VL-235B-A22B-Instruct (Yang et al., 2025) and the images, as shown in Figure 7.

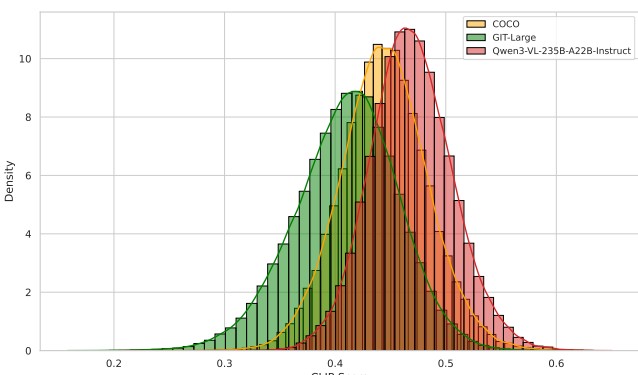

*Figure 7.* Distribution comparison of CLIP Scores across three caption sources. The density plots illustrate the semantic alignment between images and captions generated by Qwen3-VL-235B (red), GIT-large (green), and the original COCO ground truth (orange). The distinct rightward shift of the Qwen distribution indicates that our generated captions achieve superior image-text alignment, surpassing both the baseline model and the original human annotations.

## A.2. Bounds on Voxel Response Consistency

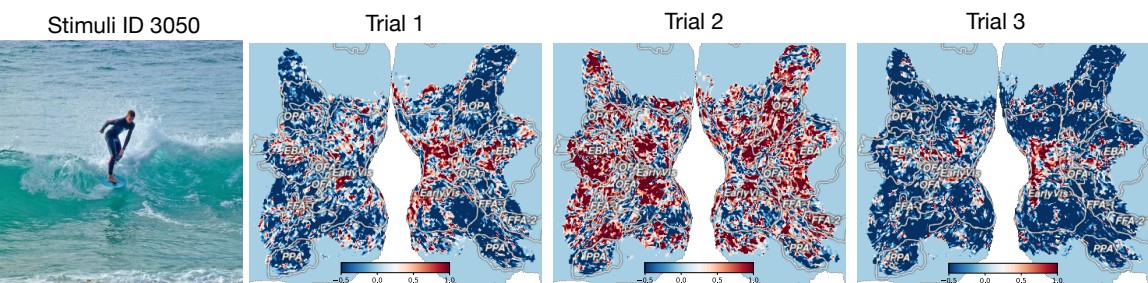

*Figure 8.* Distribution of brain activity beta values across three different trials under the same stimulus ID 3050 (Subject 1). Significant inter-trial variability can be observed.

To evaluate the reliability and intrinsic quality of the neural responses, we conducted a comprehensive voxel-level analysis across all eight subjects. We visualize brain-region activations for the same stimulus (ID 3050), as shown in Figure 8. We quantified the consistency of neural signals using three complementary metrics: Mean Squared Error (MSE), Cosine Similarity, and Pearson Correlation Coefficient (Figures 12 to 14). While MSE measures the absolute magnitude of point-wise divergence, Cosine Similarity and Pearson Correlation Coefficient capture the directional alignment and linear dependence of the voxel response vectors, respectively. The to-mean metric is computed between each individual trial and the averaged response, while the pair-wise metric is computed between all pairs of individual trials.

Crucially, these inter-trial consistency metrics establish a noise ceiling, an empirical upper bound for the performance of any image-to-voxel encoding model. Our statistical analysis indicates that due to inherent biological variability, the optimal achievable performance is bounded: the empirical lower bound for MSE is approximately 0.55, and the upper bounds for both Cosine Similarity and Pearson Correlation are below 0.65.

### A.3. Implementation details

**Environment.** Our method is implemented using Python 3.12.11, CUDA 12.8, PyTorch 2.8.0, transformers 4.57.1, and flash_attn 2.8.1, on Ubuntu 22.04.05 LTS. We use accelerate 1.10.1 with ZeRO Stage 2 and bfloat16 (BF16) precision. All experiments are conducted on a machine equipped with 96 vCPUs (2.90 GHz) from Intel Xeon processors, eight NVIDIA A100 GPUs with 80 GB memory each, and 1024 GB of RAM.

**Network architecture and training configuration.** For the brain tokenizer, we adopt a VQ-VAE architecture with a codebook size of 128, a compression ratio of 128, and an embedding dimension of 32. The tokenizer is trained using the AdamW (Loshchilov & Hutter, 2017) optimizer with an initial learning rate of $1 \times 10^{-4}$, a batch size of 256, and for 100 epochs. We set the commitment loss coefficient to 0.25 and the entropy loss ratio to 0.1. For autoregressive modeling, we employ the AdamW optimizer with a cosine learning rate schedule and warmup, where the peak learning rate is set to $2 \times 10^{-4}$. The model is trained for 15 epochs with a batch size of 16, using ZeRO Stage-2 optimization. The backbone is initialized with the parameters of Janus-7B, and the hidden dimension is set to 4096. During this stage, other components such as the VQ-VAE are frozen. All hyperparameters can be seen the table as follows Tables 6 and 7.

*Table 6.* Hyperparameters of BrainTokenizer

| Hyperparameter | Value |
|---|---|
| Base channel size | 64 |
| Encoder channel multipliers | [1, 2, 2, 2, 4, 4, 4] |
| Decoder channel multipliers | [1, 2, 2, 2, 4, 4, 4] |
| Residual blocks per level | 2 |
| Downsampling factor per level | 2 |
| Latent channel dimension | 512 |
| Codebook size | 128 |
| Codebook embedding dimension | 32 |
| Commitment loss weight | 0.25 |
| Entropy regularization | 0.1 |
| Attention | Self-attention |
| Reconstruction loss | MSE |

*Table 7.* Hyperparameters for Transformer Architecture

| Hyperparameter | Value |
|---|---|
| *Language Backbone* | |
| Backbone Type | Decoder-only Transformer |
| Layers | 30 |
| Hidden size | 4096 |
| Attention heads | 32 |
| Head dimension | 128 |
| Intermediate size | 11008 |
| Activation function | SiLU |
| Vocabulary size | 102,400 |
| Max position embeddings | 16,384 |
| RMSNorm epsilon | $1 \times 10^{-6}$ |
| RoPE $\theta$ | 10,000 |
| Attention dropout | 0.0 |
| *Multimodal Components* | |
| Aligner | depth=2, input_dim=1024, n_embed=4096 |
| Gen-vision | image_token_size=16384, n_embed=8 |
| Gen-aligner | depth=2, input_dim=8, n_embed=4096 |
| Gen-head | image_token_embed=4096, image_token_size=16384, n_embed=4096 |

# B. Detailed Results

## B.1. Detailed Results for Each Subject

The quantitative results for the brain-to-text and brain-to-image tasks are reported in Table 8 and Table 9, respectively.

*Table 8.* Quantitative evaluation of brain-to-text caption generation across all subjects.

| Method | BLEU1↑ | BLEU2↑ | BLEU3↑ | BLEU4↑ | METEOR↑ | ROUGE↑ | CIDEr↑ | SPICE↑ | BERT↑ | CLIP-Text↑ | CLIP-Image↑ |
|--------|--------|--------|--------|--------|---------|--------|--------|--------|-------|-----------|-------------|
| Subj 1 | 0.398 | 0.211 | 0.116 | 0.068 | 0.141 | 0.294 | 0.426 | 0.121 | 0.382 | 0.327 | 0.327 |
| Subj 2 | 0.400 | 0.208 | 0.113 | 0.064 | 0.138 | 0.292 | 0.391 | 0.111 | 0.377 | 0.279 | 0.287 |
| Subj 3 | 0.399 | 0.208 | 0.110 | 0.062 | 0.135 | 0.286 | 0.378 | 0.115 | 0.374 | 0.255 | 0.019 |
| Subj 4 | 0.390 | 0.202 | 0.108 | 0.061 | 0.132 | 0.282 | 0.362 | 0.108 | 0.362 | 0.229 | 0.015 |
| Subj 5 | 0.409 | 0.218 | 0.119 | 0.068 | 0.143 | 0.300 | 0.442 | 0.124 | 0.391 | 0.320 | 0.333 |
| Subj 6 | 0.401 | 0.209 | 0.113 | 0.065 | 0.140 | 0.292 | 0.400 | 0.117 | 0.379 | 0.320 | 0.027 |
| Subj 7 | 0.400 | 0.209 | 0.114 | 0.067 | 0.137 | 0.289 | 0.394 | 0.113 | 0.375 | 0.253 | 0.272 |
| Subj 8 | 0.378 | 0.188 | 0.095 | 0.052 | 0.126 | 0.271 | 0.309 | 0.092 | 0.347 | 0.157 | 0.011 |
| Avg. | 0.397 | 0.207 | 0.111 | 0.063 | 0.136 | 0.288 | 0.388 | 0.113 | 0.373 | 0.268 | 0.161 |

*Table 9.* Quantitative evaluation of brain-to-image generation across all subjects.

| Method | PixCorr↑ | SSIM↑ | Alex(2)↑ | Alex(5)↑ | Incep↑ | CLIP↑ | Eff↓ | SwAV↓ |
|--------|----------|-------|----------|----------|--------|-------|------|-------|
| Subj 1 | 0.195 | 0.293 | 91.5% | 97.2% | 95.1% | 94.9% | 0.648 | 0.361 |
| Subj 2 | 0.173 | 0.287 | 90.0% | 96.1% | 94.7% | 93.7% | 0.663 | 0.376 |
| Subj 3 | 0.157 | 0.284 | 87.7% | 95.0% | 93.0% | 93.3% | 0.698 | 0.392 |
| Subj 4 | 0.159 | 0.285 | 87.3% | 94.6% | 91.8% | 92.4% | 0.694 | 0.397 |
| Subj 5 | 0.169 | 0.286 | 89.3% | 96.5% | 95.7% | 95.6% | 0.634 | 0.361 |
| Subj 6 | 0.169 | 0.283 | 88.3% | 95.8% | 94.6% | 94.6% | 0.660 | 0.380 |
| Subj 7 | 0.154 | 0.284 | 87.1% | 95.1% | 93.5% | 93.5% | 0.680 | 0.389 |
| Subj 8 | 0.144 | 0.281 | 84.7% | 92.6% | 88.4% | 89.9% | 0.747 | 0.431 |
| Avg. | 0.165 | 0.285 | 88.2% | 95.3% | 93.3% | 93.5% | 0.678 | 0.386 |

## B.2. Brain Decoding Cases

Figure 9 and Figure 10 present qualitative results for brain decoding. For brain-to-text, the model generates coherent descriptions that capture salient objects and high-level semantics. Similarly, brain-to-image reconstructions exhibit reasonable global structure and semantic consistency. Despite some degradation in fine-grained details, the generated images correctly reflect object categories and spatial layouts, demonstrating effective decoding into both language and vision modalities.

## B.3. Brain Encoding Cases

Figure 11 illustrates brain encoding performance, including image-to-fMRI prediction and subsequent reconstruction. The synthesized fMRI responses are structurally consistent with real brain activity. Furthermore, images reconstructed from these predicted signals retain recognizable semantic content and visual characteristics. These results confirm the model's capability to capture bidirectional mappings between visual stimuli and brain activity within a unified framework.

## B.4. Visual-to-fMRI Hacking Analyse

**Voxel-level variability.** Previous encoding metrics (Gu et al., 2022; Wang et al., 2023) predominantly rely on voxel-wise metrics, such as Pearson correlation and MSE. However, these local measures suffer from two fundamental limitations. First, they neglect global cortical topography, failing to penalize structurally incoherent predictions. Second, they are overly sensitive to the intrinsic stochasticity of neural responses (i.e., trial-to-trial variability), often penalizing plausible signal variations that deviate from a specific, noisy ground-truth recording.

**Semantic-level Hacking.** To overcome the challenges of low-level variability in fMRI, recent studies (Bao et al., 2025; Mai et al., 2025) have established a semantic-level evaluation protocol. This paradigm assesses synthesized neural signals

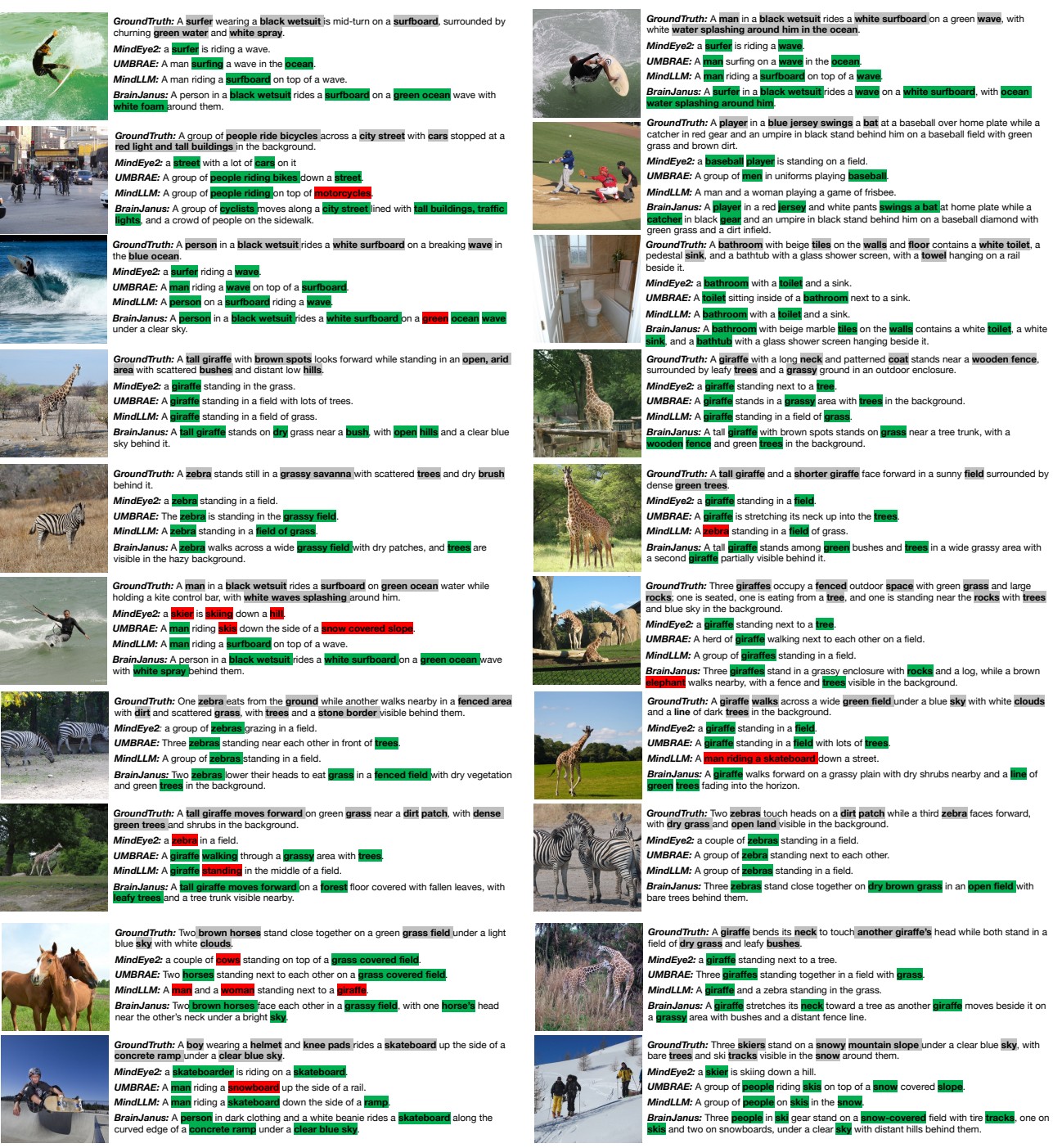

*Figure 9.* Qualitative comparison of brain caption decoding results. GroundTruth image captions are compared with captions decoded from fMRI voxel signals using MindEye2, UMBRAE, MindLLM, and BrainJanus (ours). Green highlights indicate semantic matches with the GroundTruth, while red highlights denote errors.

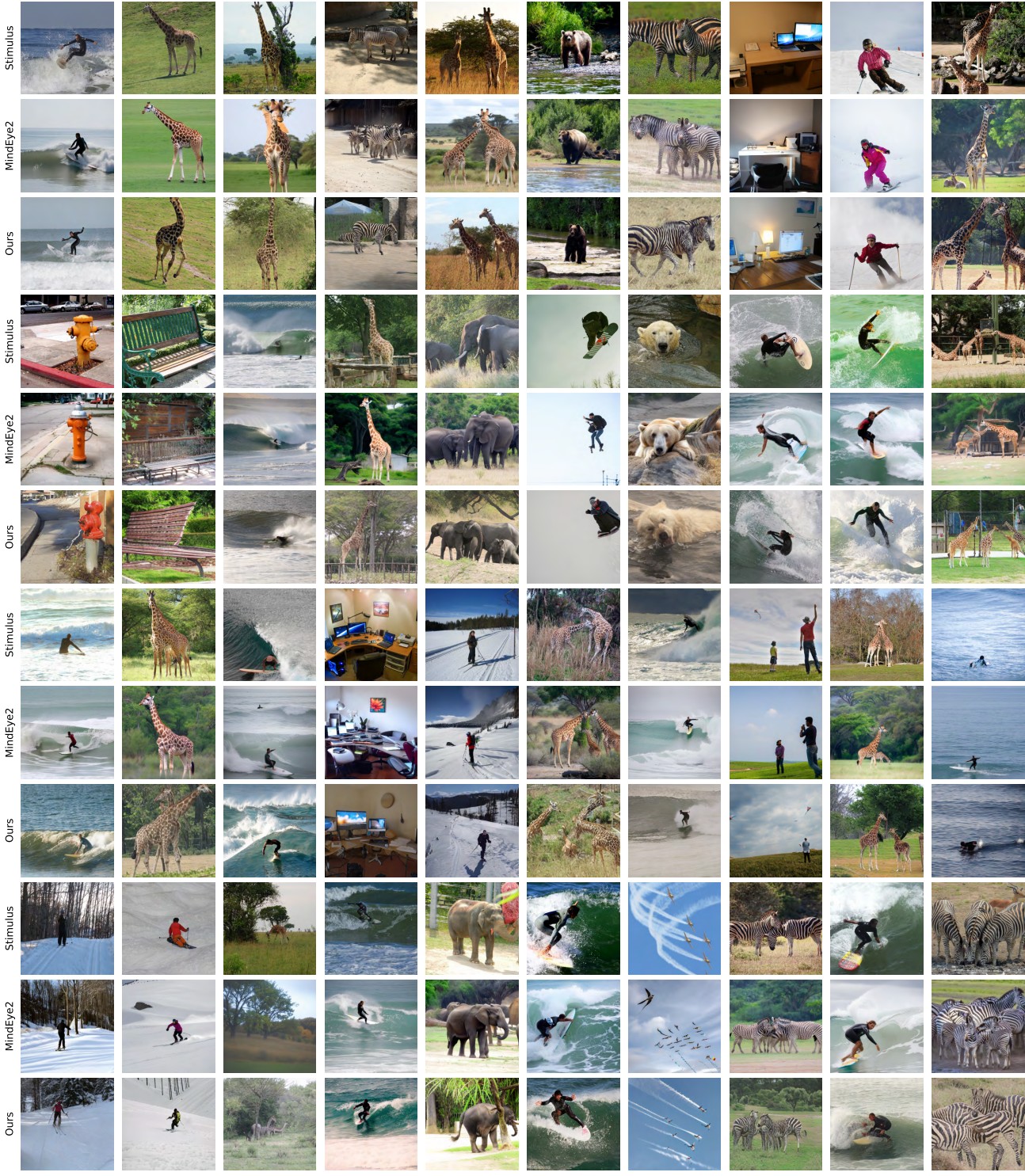

*Figure 10.* Qualitative results of brain-to-image decoding.

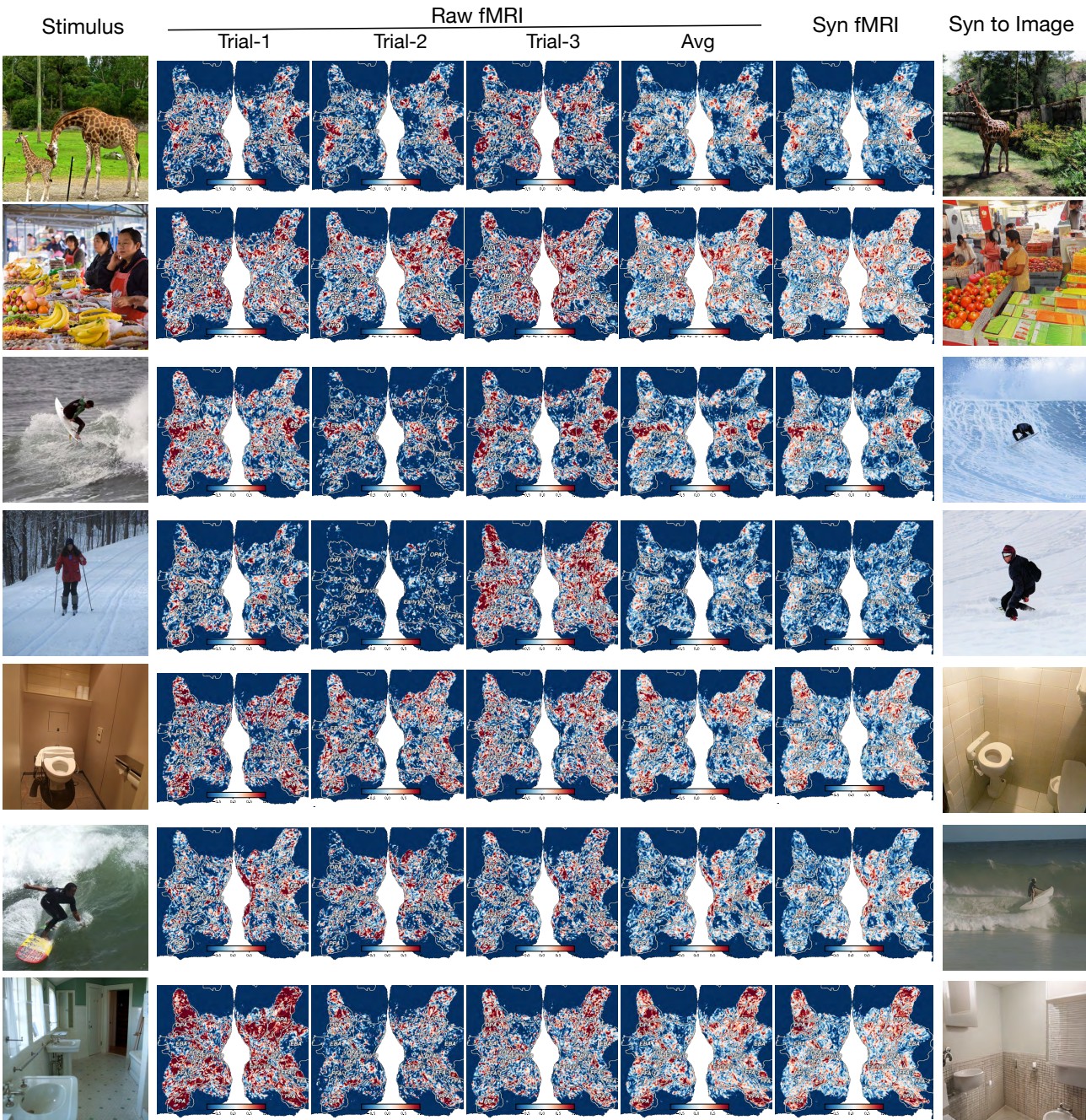

*Figure 11.* Qualitative results of image-to-brain encoding.

by measuring the fidelity of images reconstructed via a encoding-decoding pipeline (Image → Visual Embedding → Syn-fMRI → Reconstructed Image). However, we demonstrate that this protocol is susceptible to a trivial hacking strategy. We introduce a naive baseline, the Padding Hacking model, as illustrated in Figure 16. Instead of learning a biological mapping, this encoder simply zero-pads the ground-truth visual embedding (e.g., VQ-VAE or CLIP) to match the voxel dimension $N$, effectively treating the voxel space as a sparse storage container. The decoder performs the inverse un-padding operation to retrieve the embedding for generation. As shown in Table 4, this biologically invalid approach achieves perfect scores across all low-level and high-level semantic metrics. This experiment reveals a critical flaw in the evaluation protocol: it fails to distinguish between true neural synthesis and mere information leakage. Consequently, to ensure validity, training methodologies must strictly prohibit alignment with pre-trained visual embeddings.

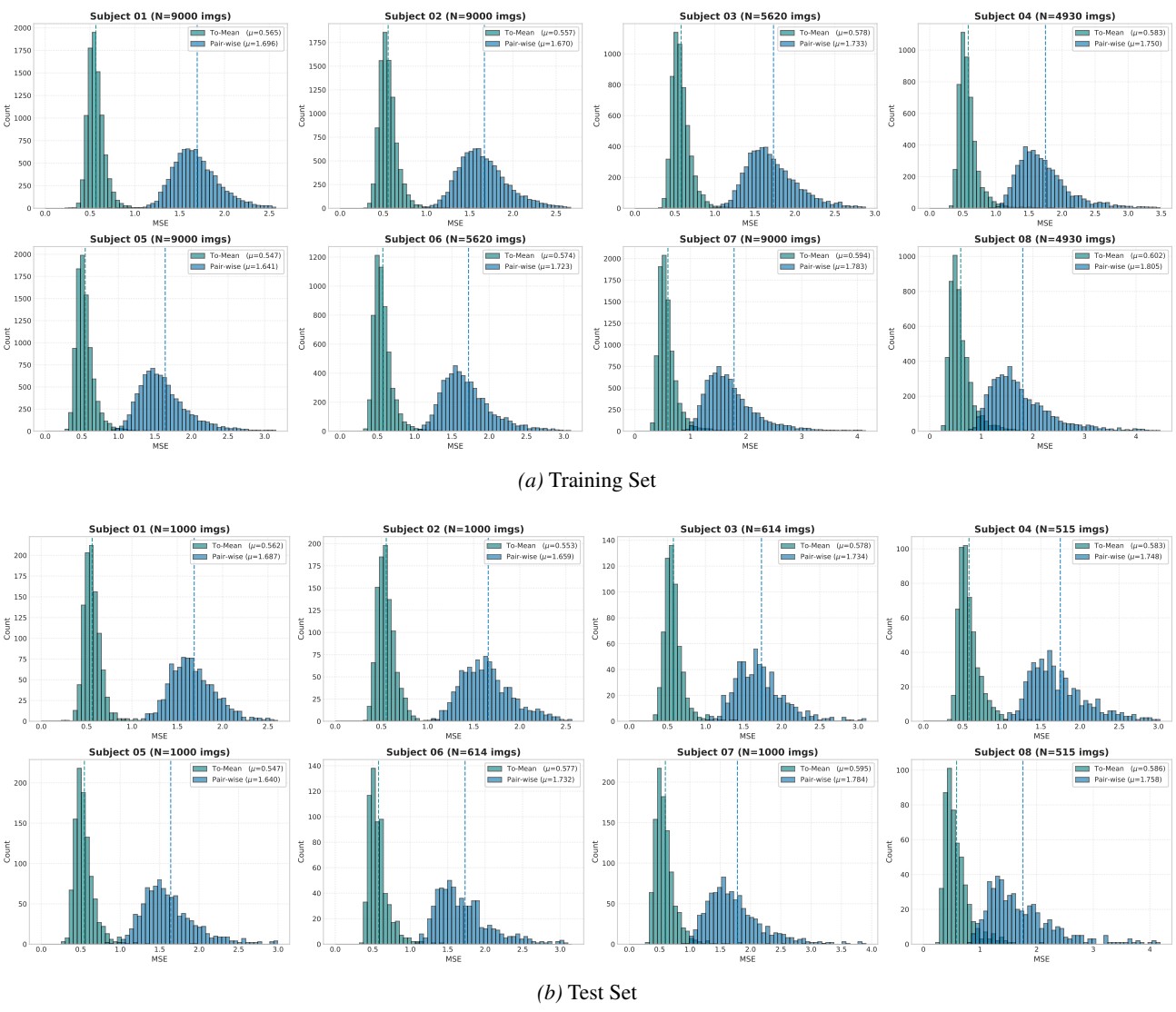

*(a)* Training Set

*(b)* Test Set

*Figure 12.* Distributions of voxel consistency measured by **Mean Squared Error (MSE)** in (a) the training set and (b) the test set. Histograms show voxel MSE values based on three repeated trials per image. **Blue (Pair-wise):** MSE computed between pairs of trials for the same stimulus. **Teal (To-Mean):** MSE between each trial and the mean response across repetitions of the same stimulus. Vertical dashed lines indicate the mean ($\mu$).

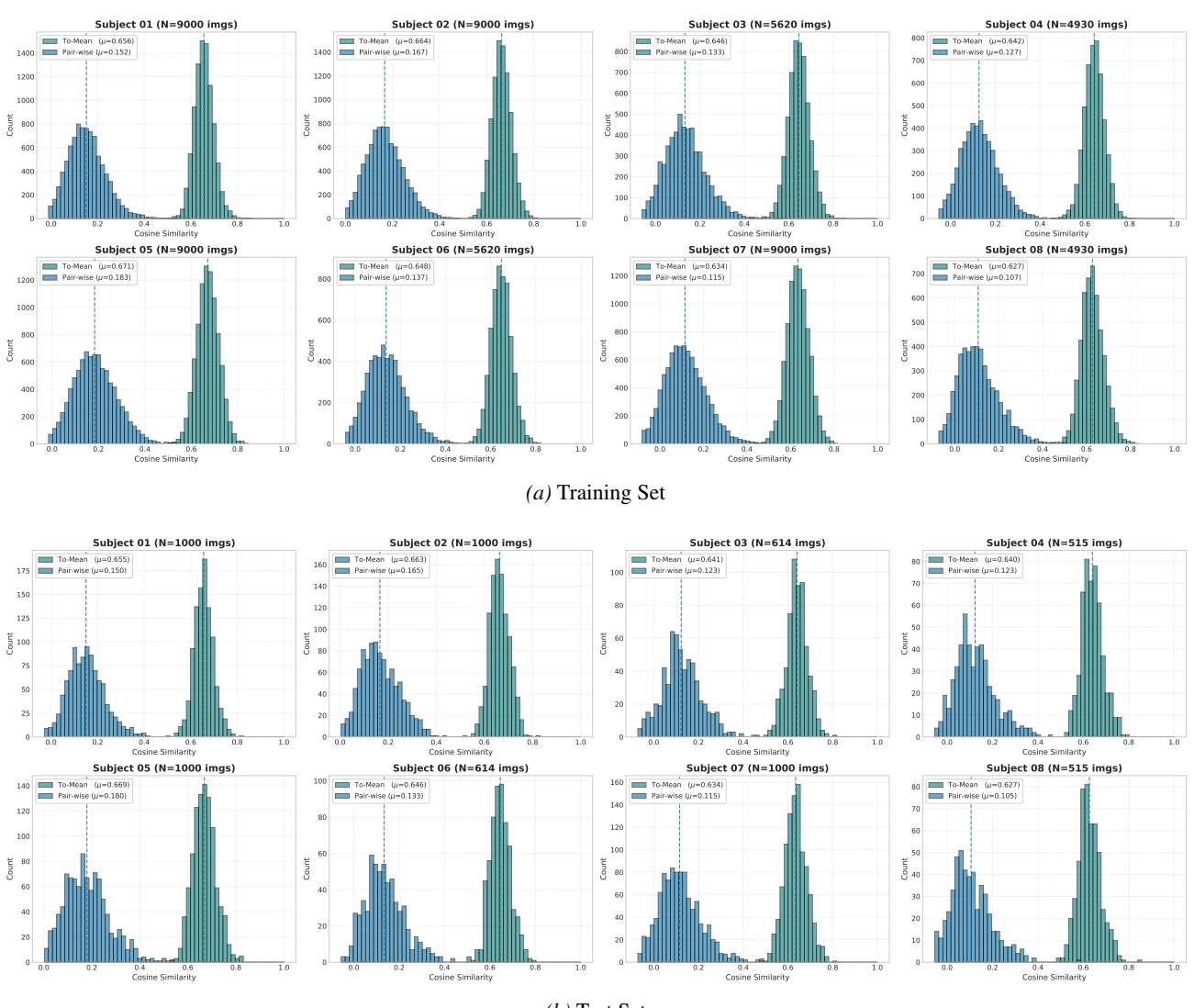

*(a)* Training Set

*(b)* Test Set

*Figure 13.* Distributions of voxel consistency measured by **Cosine Similarity.**

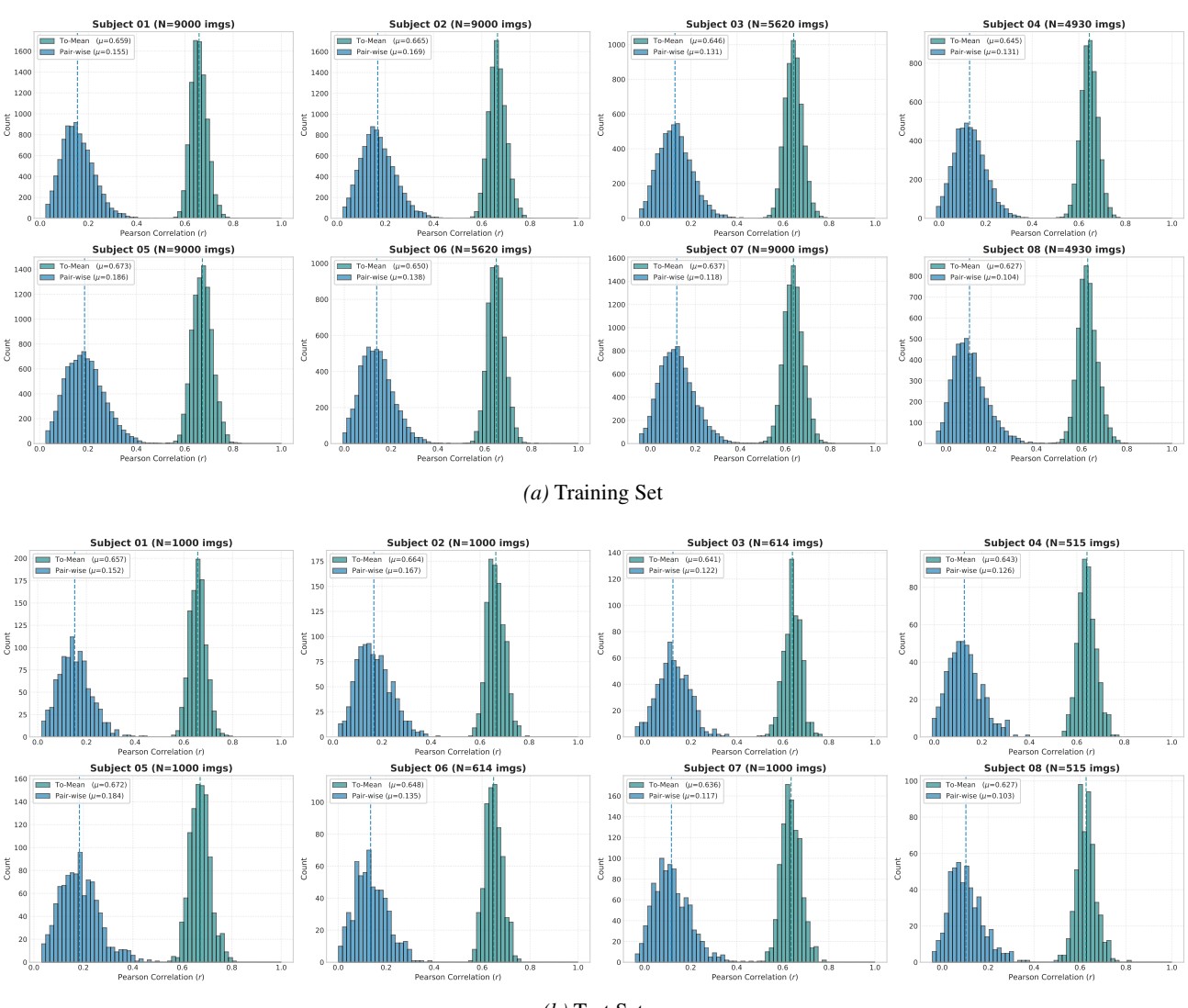

*(a)* Training Set

*(b)* Test Set

*Figure 14.* Distributions of voxel consistency measured by **Pearson Correlation Coefficient.**

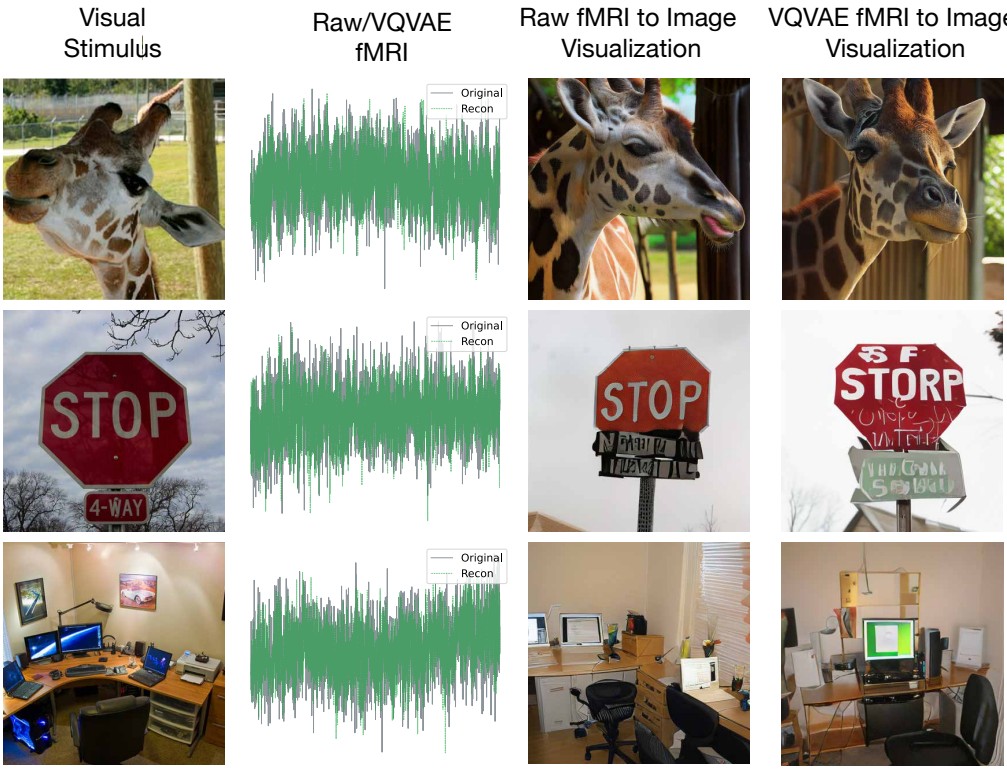

*Figure 15.* Qualitative result of Brain Tokenizer Pretraining.

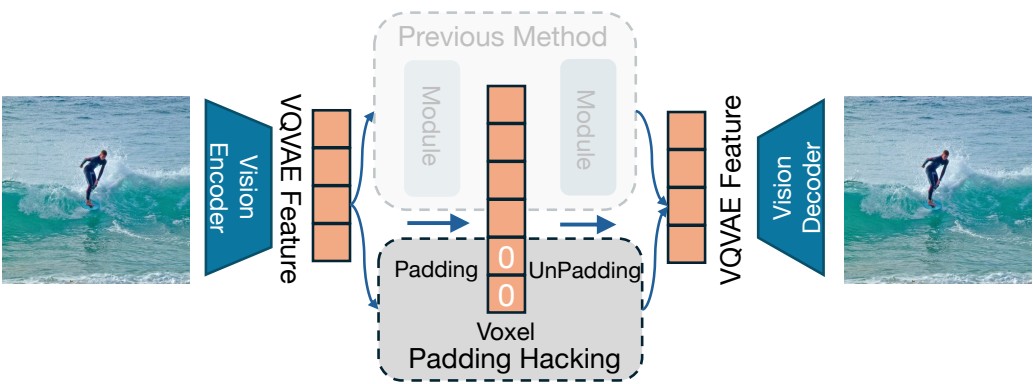

*Figure 16.* Illustration of the proposed Padding Hacking baseline under the encoding-decoding protocol. Instead of learning a biologically meaningful mapping from images to neural responses, the encoder directly stores the ground-truth visual embedding (e.g., CLIP or VQ-VAE) by zero-padding it to match the voxel dimensionality. The decoder then performs the inverse un-padding operation to recover the embedding for image reconstruction. This trivial information-leakage strategy can achieve near-perfect reconstruction scores, revealing a critical vulnerability of reconstruction-based semantic evaluation for visual-to-fMRI synthesis.

