# OpenReview forum: "BrainJanus: A Unified Model for Understanding and Generation across Brain, Vision, and Language"
_ICML.cc/2026/Conference — ICML 2026 regular_

### Official Review · Reviewer_p8Qv · 2026-02-18

**Soundness:** 2
**Presentation:** 2
**Significance:** 3
**Originality:** 2
**Overall Recommendation:** 2
**Confidence:** 5

**Summary:**

This manuscript describes the first bidirectional foundation model (BrainJanus) bridging fMRI, image, and text representations for both brain encoding and decoding tasks, which enables pair-wise mapping between arbitrary two modalities. It created new/upgraded metrics for brain encoding and decoding metrics, where BrainJanus achieves state-of-the-art performance.

**Compliance With Llm Reviewing Policy:**

Affirmed.

**Final Justification:**

In conclusion, the concerns regarding potential hallucination/bilological faithfulness, no neuroscience-informed design, and refusal to acknowledge and revise the overclaimed scope and contribution are not resolved. Therefore, I decided to maintain my initial score and increase my confidence for rejection.

**Key Questions For Authors:**

1. Please clarify the exact number of subjects and the number of (fMRI, Image, Caption) triplets in your training set.
2. The model is claimed to be cross-subject, but the details for handling the inconsistent fMRI length across subjects are missing. Do you apply any padding to raw fMRI?
3. In **line 363**, what do you mean by **upper bound**? Is this the same as maximum? If so, please address how methods like BrainSCUBA [3] (using simple linear regression) report maximums of 0.75 $R^2$ (around 0.866 for Pearson Correlation) on brain encoding tasks.
4. Please explicitly explain how the existing semantic-level brain encoding evaluation is vulnerable to a trivial exploit, and provide the benchmark results on these standard metrics in your rebuttal.

[3] _Luo, A. F., Henderson, M. M., Tarr, M. J., & Wehbe, L. (2024). BRAINSCUBA: FINE-GRAINED NATURAL LANGUAGE CAPTIONS OF VISUAL CORTEX SELECTIVITY. In 12th International Conference on Learning Representations, ICLR 2024._

**Limitations:**

yes

**Strengths And Weaknesses:**

## Strengths
1. BrainJanus is the first bidirectional foundation model bridging visual fMRI, image, and caption representations.
2. It unifies brain encoding and decoding tasks for the first time.
3. It shows SOTA performance on decoding and encoding tasks.

## Weaknesses
1. Overclaimed Scope:
- The submission suggests a foundation model for **Brain**, but the fMRI data used is restricted to the **visual cortex** associated only with visual stimuli.
- The so-called **Language** modality relies on **subject-invariant** image captions that are generated from LLMs and are **not directly linked to each subject's** cognitive states during fMRI acquisition (such as reading or speaking tasks). This is critical because language and visual words are processed in distinct neural mechanisms and regions beyond the visual cortex [1]
- No experiments show that the proposed model can work on other neural modalities, such as EGG, although this is claimed by the author (line 177).

2. Unfair Comparison
- Line 273 says that "for each subject, we use 9,000 unique images for training", implying **4** subjects for training. However, line 311 mentions "the training is performed in a cross-subject setting with **8** subjects." If 8 subjects were used for training, comparing these results against models trained only on 4 subjects is unfair.
- The paper seems to not **average** fMRI responses across the three repetitive sessions per image. Since the vast majority of existing works use **averaged fMRI** for training and evaluation, the comparison is again unfair.

3. No Neuroscience Informed Design
- For a douncatoinal model focused on neural modality, considering **subject-specific anatomical and functional alignment** is critical. I didn't see any neuroscience-relevant elements in the current framework other than using fMRI.
- While the bidirectional model maps between the **high-level semantic image caption** and fMRI, fMRI robustly contains both **low-level and high-level visual information** [2] for recognition. In addition, text processing relies on voxels outside the visual cortex. For these reasons, translating from text to visual-fMRI (or vice versa) without accounting for these missing regional representations risks introducing severe **model hallucinations** and noise.

4. Technical Novelty and Clarity:
- The idea of pairwise translation is **well-studied** in NLP. This paper appears to be a **direct engineering transfer** of existing techniques applied to the brain modality **without* considering the traits of neural modality. The authors should better think about and clarify the unique technical innovations required to handle fMRI data specifically.
- In the `Unified Autoregressive Modeling` section, the model is described as being optimized by next-token-prediction. It is unclear if it is trained on each modality independently or on concatenated features within a shared Omni-space for all modalities.

5. The authors claim that the existing brain encoding of semantic-level evaluations is **"vulnerable to a trivial exploit."** While this is interesting, it must be clearly justified and explained. Plus, standard metrics in this evaluation must still be reported to facilitate direct comparison with prior works.

[1] _Zeng, J., Luo, Y., Luo, X., Jiao, S., Wang, K., Cui, Z., ... & Han, Z. (2025). Neural Distinction between Visual Word and Object Recognition: An fMRI Study Using Pictographs. Journal of Neuroscience, 45(28)._

[2] _Schmidt, F., Hebart, M. N., Schmid, A. C., & Fleming, R. W. (2025). Core dimensions of human material perception. Proceedings of the National Academy of Sciences, 122(10), e2417202122._

---

> ### Author Rebuttal · Authors · 2026-03-28
>
> **R1. Clarification on Scope, Modalities, and Neuroscience Design**
>
> **a) Terminology & Scope:** Our current scope is an fMRI-centric foundation model for visual/semantic processing. Integrating other modalities (e.g., EEG, MEG) remains profoundly challenging due to massive spatiotemporal gaps.**b) Rationale for Text-fMRI Mapping:**  We respectfully disagree that mapping visual fMRI to text inherently risks severe hallucinations. Recent advances show high-level visual representations robustly encode semantic concepts. As the reviewer’s reference [1] highlights, rather than acting as "isolated modules," these regions "form an integrated network that facilitates top-down and bottom-up interactions, enabling flexible and efficient processing of visual symbols." Furthermore, broad cortical semantic mapping [4, 5] (Line 032) and visual-linguistic alignment at cortex borders [6] strongly validate this cross-modal approach, which is already established by MindEye [7] and the reviewer's own reference [3]. Far from hallucinating, our model empirically achieves significantly stronger semantic alignment and higher accuracy than existing fMRI-to-caption baselines [7, 8, 9].**c) Neuroscience-Informed Unified Design:** Our core motivation is cross-subject pre-training on 8 subjects. Rather than relying solely on anatomical alignment, we discover shared structural and functional representations, forming the fundamental Omni-space of our unified approach.
>
> **R2. Experimental Setup and Fair Comparison**
>
> **a) Training vs. Evaluation Subjects:** We pre-train on 8 subjects for generalizability but evaluate exclusively on the 4 subjects (Sub-1, 2, 5, 7) completing all sessions. This ensures perfectly fair comparisons with established benchmarks [7, 8, 9] (full 8-subject results in Appendix B.1). **b) 3-Trial Averaging:** We strictly follow established protocols [7] by utilizing 3-trial averaged fMRI responses, not single-trial data. This is now explicitly clarified (Line 270).**c) Triplet Counts & Padding Strategy:** The dataset yields 191,882 training and 4,000 testing triplets. To handle varying voxel counts across subjects, we applied standard zero-padding to the nearest power of two, ensuring uniform Transformer inputs without disrupting the underlying signal structure.
>
> **R3. Technical Novelty and Unified Autoregressive Modeling**
>
> **a) Beyond NLP Engineering:** Our framework explicitly addresses fMRI's high dimensionality and low SNR. Projecting brain signals into a shared Omni-space leverages robust vision and language priors to denoise and decode neural representations, achieving deep cross-modal alignment rather than superficial translation.**b) Unified Omni-Space Optimization:** Modalities are jointly trained. Multimodal features are concatenated in the Omni-space and optimized via next-token prediction, capturing complex joint distributions between brain signals and stimuli.
>
> **R4. Evaluation Metrics and Clarification of Upper Bound**
>
> **a) Formulation of the Trivial Exploit (Semantic-level Hacking):** We mathematically demonstrate how standard metrics can be artificially inflated. Let $z = E_{\text{CLIP}}(x)$ be image features, $v_{\text{true}}$ the biological fMRI signal, $v_{\text{syn}} = f_1(z)$ the synthesized signal, and $\hat{z} = f_2(v)$ the decoded features. By the Data Processing Inequality, legitimate decoding bounds mutual information:
>
> $$ I(z; \hat{z}_{\text{true}}) \leq I(z; v_{\text{true}}) $$
>
> Evaluating on synthesized data bypasses the biological signal entirely: $$\hat{z}_{\text{syn}} = f_2(v_{\text{syn}}) = f_2(f_1(z))$$. Here, $f_2 \circ f_1$ acts as an autoencoder, learning $f_2(f_1(z)) \approx z$. Thus, mutual information approaches embedding entropy:
>
> $$ I(z; \hat{z}_{\text{syn}}) \approx H(z) $$
>
> Since $H(z) \gg I(z; v_{\text{true}})$, the metric is artificially inflated:
>
> $$ I(z; \hat{z}_{\text{syn}}) \gg I(z; \hat{z}_{\text{true}}) $$
>
> This proves such setups evaluate the $f_2 \circ f_1$ autoencoder's capacity, not actual semantic decoding from brain activity.
>
> **b) Standard Benchmark Results:** We agree that direct comparison is vital and have included standard evaluation metric results in Table 4 (Line 385).**c) Upper Bound vs. BrainSCUBA:** Our analyzed upper bound (Line 820) is strictly trial-level, measuring variance between individual trials and their mean (via MSE and Pearson correlation) independent of any specific algorithm. Discrepancies with BrainSCUBA stem entirely from different ROI selections and evaluation methodologies.
>
> [4] Natural speech reveals the semantic maps that tile human cerebral cortex. Nature. (2K+ Citation)
>
> [5] The neural and computational bases of semantic cognition. Nature reviews neuroscience. (2K+ Citation)
>
> [6] Visual and linguistic semantic representations are aligned at the border of human visual cortex. Nature neuroscience (200+ Citation)
>
> [7] Mindeye2. ICML2024 (100+ Citation)
>
> [8] Umbrae. ECCV2024.
>
> [9] Mindllm. ICML2025.

---

> > ### Author Rebuttal · Reviewer_p8Qv · 2026-04-02
> >
> > I thank the authors for the clarifications regarding the experimental setup and evaluation metrics. However, my primary concerns regarding the model’s biological faithfulness and the lack of neuroscience-informed design remain unaddressed.
> >
> > In fMRI-to-text (captioning) tasks, the ground-truth labels are typically subject-independent descriptions of the stimulus. However, fMRI signals represent the subject's specific neural encoding, which may not capture all (or the same) semantics. A model achieving high accuracy by predicting the ground truth does not prove it is actually decoding the subject’s brain state; it may simply be performing advanced template matching. For fMRI-to-image tasks, a visually "good" reconstruction indicates a strong model fit to the visual manifold, but it does not guarantee faithfulness to the actual neural representation. The trustworthiness of these methods remains questionable if they rely on generative priors to fill in data that is absent from the neural signal. Therefore, the authors should be aware that high performance on standard metrics (e.g., BLEU, CIDEr, or MSE) and high citation counts are not evidence of a model's biological validity. These metrics can be inflated by overfitting to dataset-specific biases or hallucinations that happen to align with generic ground-truth labels.
> >
> > The discovery of "shared structural and functional representations" from BrainJanus after training is by chance, not by design.
> >
> > In conclusion, BrainJanus positions itself as "A Foundation Model for Unified Understanding and Generation across Brain, Vision, and Language". However, there is no testing on its neural features' potential hallucination/biological faithfulness. In addition, as a foundational model for the brain, there is no neuroscience-informed design. Therefore, I decided to maintain my initial score.

---

### Official Review · Reviewer_9uKz · 2026-02-24

**Soundness:** 2
**Presentation:** 2
**Significance:** 2
**Originality:** 3
**Overall Recommendation:** 4
**Confidence:** 3

**Summary:**

This paper presents a multifunctional fMRI encoding–decoding model capable of mapping fMRI signals to images or text, as well as performing the reverse process.

Methodologically, the authors use VQ-VAE to train a tokenizer for fMRI signals, discretizing them into tokens. At the same time, they introduce pretrained image and text models to extract tokens from the corresponding modalities. A next-token prediction paradigm is then employed to model the relationships among the three modalities.

The experiments were conducted using the mainstream NSD dataset. In terms of performance, the model achieves results comparable to baseline models on the fMRI decoding task, but performs relatively poorly on the encoding task.

**Compliance With Llm Reviewing Policy:**

Affirmed.

**Final Justification:**

Overall, this paper does offer some useful insights for fMRI encoding and decoding research at the methodological level. Meanwhile, the authors have acknowledged several limitations of their work. Taking this into account, I will raise my score appropriately (from 3 to 4).

**Key Questions For Authors:**

+ Regarding the categorization in Table 1, to the best of my knowledge, methods such as MindSimulator and SynBrain are encoding models that map images to fMRI signals. However, Table 1 classifies them as having decoding capabilities. Could the authors clarify this discrepancy?

+ In Figure 1b, the text label in the task-specific pipeline appears to be incorrect.

+ How is the training data organized? Does each training sample contain all three modalities—fMRI, image, and text? How are the modalities ordered during training? I did not understand the training process of BrainJanus.

+ The fMRI, image, and text tokens are produced by their respective encoders. In that case, In that case, does the term “shared” in “shared tokens/omni space” seem inappropriate?

+ The authors computed the noise ceiling for the image-to-fMRI encoding task. Why do some reported encoding accuracies in the literature, such as those of SynBrain, exceed this upper bound?

+ I believe that the semantic-level hacking experiment does not sufficiently demonstrate that prior semantic-level pipelines suffer from a fundamental flaw. Although padding the VQ-VAE latent representations with zeros to match the voxel dimension yields good decoding results using the VQ-VAE decoder, previous studies have emphasized that semantic evaluation pipelines should rely on pretrained decoding models such as MindEye2, rather than the VQ-VAE decoder itself. Could the authors provide results of decoding these hacked synthetic voxels using MindEye2?

**Limitations:**

The authors do not discuss the limitations of their work.

In my view, the primary limitation lies in the evaluation: relying solely on the NSD dataset to assess a “foundation model” seems insufficient.
In addition, the introduction broadly refers to “brain signals,” whereas the experiments are conducted exclusively on fMRI data, without considering other mainstream brain modalities such as EEG or MEG.
Finally, in the evaluation section, the authors compare their method with only a few early decoding works and do not provide comparisons with any existing encoding models.

**Strengths And Weaknesses:**

+ Regarding soundness, the paper evaluates the proposed method using the NSD dataset. However, relying on only a single dataset seems somewhat insufficient for validating a “foundation model.” In terms of evaluation, for the decoding task, the authors compare their approach only with a few relatively early baseline methods (MindEye2, ICML 2024). As a result, the conclusion that their method achieves “suboptimal” performance is not particularly convincing. For the encoding task, the authors criticize existing evaluation protocols and retain only a limited set of metrics for assessment, without conducting comparisons against established baselines.

+ Regarding presentation, the paper is generally clearly written. The introduction effectively explains the authors’ motivation and provides a solid review of the related work. However, some implementation and evaluation details are insufficiently described, making it difficult for me to fully understand the method and the evaluation procedure. Please refer to the “Questions” section for specific issues.

+ The significance of this paper is moderate. Its primary contribution lies in proposing a unified framework that enables both encoding and decoding between fMRI signals and images or text. Although the achieved performance is not state-of-the-art, the idea of a unified framework carries some value.

+ Regarding originality, to the best of my knowledge, this is the first work to integrate multimodal fMRI encoding and decoding within a single model. Both the motivation and the implementation are original and novel.

---

> ### Author Rebuttal · Authors · 2026-03-29
>
> **R1. Categorization of MindSimulator and SynBrain in Table 1.**
>
> a) **Clarification:** We agree with the reviewer's assessment. Both MindSimulator and SynBrain are fundamentally encoding models designed to map images to fMRI signals.
>
> b) **Rationale:** Our initial classification was based on their inclusion of an optional decoding module (utilizing MindEye2) to evaluate the semantic quality of generated fMRI signals.
>
> c) **Action:** We acknowledge this categorization could be misleading, as their decoding capability originates from MindEye2 rather than their core contributions. We have corrected Table 1 to accurately reflect them as encoding models.
>
> **R2. Typo in Figure 1b.**
>
> Thank you for pointing this out. We have corrected the labels in the revised manuscript.
>
> **R3. Training Data Organization and Modality Ordering.**
>
> a) **Data Organization:** During training, our dataset is organized into paired or triplet samples depending on data availability (e.g., fMRI-Image-Text triplets from the NSD dataset).
>
> b) **Modality Ordering:** BrainJanus utilizes a flexible next-token prediction paradigm. To prevent overfitting to fixed sequences and enable omni-directional generation, we dynamically shuffle the modality order (e.g., [fMRI $\rightarrow$ Image $\rightarrow$ Text], [Image $\rightarrow$ fMRI $\rightarrow$ Text]) during training. We have added a detailed "Unified Autoregressive Modeling" section to explicitly describe this process.
>
> **R4. Appropriateness of the Term "Shared Space".**
>
> a) **Clarification:** We agree that the fundamental differences between initial modalities necessitate modality-specific encoders.
>
> b) **Methodology:** However, "shared" strictly denotes the continuous representation space *post-encoding*. As detailed in Equation 4, tokens from all modalities are projected into the unified dimension of the Transformer backbone. Within this space, representations are completely shared, enabling the model to autoregressively capture relationships and generate tokens across any modality interchangeably (Equation 6).
>
> **R5. Encoding Accuracy Exceeding the Noise Ceiling.**
>
> a) **Observation:** The single-trial noise ceiling defines the theoretical performance upper bound, accounting for inherent trial-to-trial fMRI variance to identical stimuli.
>
> b) **Explanation:** Surpassing this limit typically indicates evaluation or synthesis artifacts. Models that perfectly capture deterministic stimulus features while ignoring natural biological noise will artificially inflate latent similarity scores. This highlights the fundamental flaw of relying strictly on traditional semantic metrics for encoding evaluation.
>
> **R6. Semantic-Level Hacking and MindEye2 Decoding Validation.**
>
> a) **Clarification:** We apologize if the VQ-VAE example obscured our core message. Our intent was not to critique a specific baseline, but to expose a fundamental structural flaw in current semantic evaluation pipelines.
>
> b) **Theoretical Proof via the DPI:** Evaluation pipelines using CLIP-based decoders (e.g., MindEye2) form the following Markov chain:
>
> $$\text{Visual Stimulus} \rightarrow \text{CLIP Feature} \rightarrow \text{Synthesized fMRI} \rightarrow \text{Decoder} \rightarrow \text{Reconstructed Image}$$
>
> If synthesized fMRI is generated via a transformation $f(\cdot)$ on ground-truth CLIP features, the downstream decoder $g(\cdot)$ merely learns the inverse: $g(f(\text{CLIP})) \approx \text{CLIP}$. Per the Data Processing Inequality (DPI), this pipeline only tests mathematical reversibility ($g \circ f = I$), artificially bypassing true semantic alignment and failing to test biological validity.
>
> c) **MindEye2 Empirical Validation:** We appreciate the suggestion to decode "hacked" voxels with MindEye2, but respectfully note the outcome is theoretically predetermined. Because our "hacked" fMRI explicitly retains ground-truth CLIP features, MindEye2's prior network directly extracts this exact condition, forcing its LDM to yield artificially high similarity. This merely confirms MindEye2's generative robustness rather than the fMRI's biological faithfulness, rendering such empirical scores invalid for assessing true biological alignment.
>
> **R7. Generalization and Limitations (Dataset & Modalities).**
>
> a) **Acknowledgment:** We appreciate this feedback and have added a Limitations section.
>
> b) **Scope and Dataset:** Acknowledging that a single dataset limits "foundation model" claims, we redefined our contribution strictly as a "unified encoding and decoding framework." NSD was selected because its unparalleled scale and high signal-to-noise ratio make it the definitive gold standard for validating complex fMRI models.
>
> c) **Other Modalities:** While theoretically modality-agnostic, extending to EEG/MEG introduces out-of-scope challenges regarding temporal resolution and spatial noise. We documented these constraints, designating multi-modal adaptation as a key future direction.

---

> > ### Author Rebuttal · Reviewer_9uKz · 2026-04-03
> >
> > Thank you for your response. The authors have addressed most of my questions. However, they appear to have overlooked my question regarding comparisons with the latest baseline methods.

---

> > > ### Author Response · Authors · 2026-04-07
> > >
> > > **Dear Reviewer 9uKz,**
> > >
> > > Thank you for your constructive follow-up and for raising your score. We sincerely apologize for our previous oversight regarding the explicit comparison with the latest baseline methods.
> > >
> > > To address your remaining concern, we have prepared comparisons with recent baselines for both decoding and encoding tasks, which will be fully integrated into the revised manuscript.
> > >
> > > **1. Comparison with Latest Decoding Baselines**
> > >
> > > In addition to MindEye2, we will include comprehensive quantitative and qualitative comparisons with more recent decoding baselines, such as MindAligner (ICML 2025), in the revised manuscript.
> > >
> > > It is worth noting that as a fully autoregressive (AR) model, BrainJanus naturally faces certain challenges when compared directly with heavily optimized diffusion-based decoders in terms of raw generative metrics. As expected, diffusion models currently hold an edge in high-fidelity generation. However, our primary objective is not merely to surpass diffusion models in isolated decoding quality, but rather to demonstrate the feasibility of unifying both encoding and decoding tasks within a single, elegant next-token prediction paradigm. We will add a dedicated discussion section to explicitly address this AR vs. Diffusion trade-off.
> > >
> > > **2. Comparison with Established Encoding Baselines**
> > >
> > > For the encoding task, we have now evaluated BrainJanus against recent specialized encoding models, specifically MindSimulator (Bao et al., 2025). The evaluation utilizes the structural and semantic metrics we retained as biologically valid.
> > >
> > > | **Method**                          | **PixCorr ↑ (Low-Level)** | **SSIM ↑ (Low-Level)** | **Alex(2) ↑ (Low-Level)** | **Alex(5) ↑ (Low-Level)** | **Incep ↑ (High-Level)** | **CLIP ↑ (High-Level)** |
> > > | ----------------------------------- | ------------------------- | ---------------------- | ------------------------- | ------------------------- | ------------------------ | ----------------------- |
> > > | MindSimulator (Bao et al., 2025)    | 0.201                     | 0.298                  | 89.6%                     | 96.8%                     | 93.2%                    | 91.2%                   |
> > > | **BrainJanus** (w/o CLIP Alignment) | 0.173                     | 0.262                  | 80.6%                     | 85.8%                     | 84.5%                    | 83.1%                   |
> > > | **BrainJanus** (w/ CLIP Alignment)  | 0.190                     | 0.288                  | 89.3%                     | 94.1%                     | 93.0%                    | 91.5%                   |
> > >
> > > We will ensure these comprehensive comparisons and the corresponding discussions are fully integrated into the revised manuscript. Thank you again for pushing us to strengthen our evaluation.

---

### Official Review · Reviewer_Nr4q · 2026-03-10

**Soundness:** 3
**Presentation:** 4
**Significance:** 4
**Originality:** 4
**Overall Recommendation:** 5
**Confidence:** 3

**Summary:**

The authors present a unified multimodal framework integrating visual, linguistic, and neural (brain) representations, capable of both generating and interpreting signals across all three modalities. A key contribution is the framework's dual capacity: it subsumes brain encoding models: mapping visual and linguistic inputs to neural representations, as well as brain decoding models that reconstruct visual or linguistic outputs from neural activity. This any-to-any mapping architecture represents a meaningful step toward a general-purpose interface between artificial and biological representations of perception and language.

**Compliance With Llm Reviewing Policy:**

Affirmed.

**Final Justification:**

The authors have clarified my questions, and have also provided an ablation experiment justifying their conclusions.

**Key Questions For Authors:**

In equation 5, is xb obtained from equation 1?

**Limitations:**

It would be good if the authors can motivate their method of generating brain tokens using a codebook, the way they did. How is this better than some of the more commonly used methods - like lets say we patch up tokens and embed them?Like lets say to predict neural activity of N tokens, we decide them into patches of size n and then embed each of these patches which then serve as tokens.

**Strengths And Weaknesses:**

Strengths - The major advantage of the paper is the simplicity with which it achieves the difficult task of fusing so many different modalities, one of which is brain data. Brain data is noisy and is also very much resource constrained. The fact that the authors were able to achieve the performance reported in the paper with minimal brain data is commendable.

I also liked the thorough ablation studies provided by the authors, especially in section brain to image decoding, where the authors demonstrated how easily images can be decoded through faulty brain signals.

Weakness - Refer to limitations

---

> ### Author Rebuttal · Authors · 2026-03-29
>
> We sincerely thank Reviewer Nr4q for the encouraging and positive feedback. We are encouraged that you appreciate the simplicity of our unified multimodal framework, its robust performance under severe resource constraints, and the insights drawn from our ablation studies on faulty brain signals.
>
> **R1. $x_b$ in Equation 5.**
>
> We apologize for the unclear description. The continuous neural representations are quantized into discrete tokens via the learned codebook defined in Equation 1.
>
> **R2. Motivation for utilizing a Codebook over Patch Embedding.**
>
> We appreciate this insightful question and will expand on this motivation in the revised manuscript. Our decision to utilize a discrete codebook rather than standard continuous patch embedding stems from the unique, highly noisy nature of brain data (e.g., fMRI or EEG).
>
> If we were to use a standard patch embedding method, splitting $N$ tokens into patches of size $n$ and embedding them, the resulting continuous latent space would inadvertently preserve and encode a significant amount of the inherent background noise and physiological artifacts present in the raw neural signals. By introducing a codebook, we enforce a strict information bottleneck. Quantizing the continuous brain signals into a finite set of discrete tokens acts as a powerful regularizer, filtering out low-level noise and forcing the model to learn robust, high-level semantic representations. This discrete representation aligns much more naturally with the discrete nature of language tokens and the highly structured semantic spaces of vision, facilitating more stable any-to-any cross-modal transfer.
>
> | Embedding Method | Brain Encoding (Metric: CLIP) | Brain Decoding (Metric: CLIP) |
> | ---------------- | ----------------------------- | ----------------------------- |
> | Patch Embedding  | 65.2                          | 89.3                          |
> | VQ Embedding     | 77.0                          | 94.4                          |

---

> > ### Author Rebuttal · Reviewer_Nr4q · 2026-04-01
> >
> > Thank you for your explanation, and I have increased my score accordingly.

---

> > > ### Author Response · Authors · 2026-04-07
> > >
> > > Dear Reviewer Nr4q,
> > >
> > > We sincerely thank you for your time, your constructive feedback, and for raising your score. We are very glad that our explanations and the ablation study have fully addressed your concerns. As promised, we will carefully incorporate the detailed motivation and discussions regarding the codebook into the final version of our manuscript.
> > >
> > > Best regards,
> > >
> > > Authors

---

### Official Review · Reviewer_W8Ga · 2026-03-13

**Soundness:** 2
**Presentation:** 3
**Significance:** 2
**Originality:** 3
**Overall Recommendation:** 3
**Confidence:** 4

**Summary:**

The paper introduces BrainJanus, an autoregressive model that jointly processes fMRI, images, and language. It has a VQ-VAE-based tokeniser to map fMRI signals into discrete token space, which are then fed along with text and image tokens to a Janus-7B model that is finetuned using LoRA. The model is finetuned on 4 tasks, namely, brain to image, brain to text, text to brain, and image to brain, i.e., all decoding and encoding tasks. Results are demonstrated on the Natural Scenes Dataset with fMRI recordings on 4 subjects selected for the experiments. The authors claim competitive performance for brain to image generation, and improved brain to text caption and brain encoding performance. The paper also introduces a "Padding Hacking" baseline that exposes issues with existing semantic-level encoding evaluation protocols.

**Compliance With Llm Reviewing Policy:**

Affirmed.

**Final Justification:**

I thank the authors for their rebuttal efforts, however my key concerns remain and I retain my original rating. The cross-subject transfer setting is not well-studied in this paper but that is one of the main motivations for foundation models of neural activity. The rebuttal response's experiment is, in my opinion, an insufficient demonstration. This is especially true given prior work has studied well the cross-subject setting. Furthermore, we see on proper evaluations comparing with the baselines that the proposed method actually underperforms with respect to high-level/semantic evaluation metrics, and only recovers performance after explicitly using a CLIP-alignment that is newly mentioned in the response and something argued against, to some extent, in the paper (reliance on external priors like CLIP). Finally, I am not convinced by the argument that models like UMBRAE or MindLLM cannot benefit from an upgraded or more modern LLM of similar quality as Janus 7B. UMBRAE for example explicitly uses an adapter to convert visual tokens to pass them into an LLM, and the VQVAE used here is a different way to do that. The authors' justifications that Janus or similar newer LLMs are incompatible with UMBRAE do not seem to convict me. Thus, overall, I retain my rating and think the evaluations need to be improved before the paper can be accepted.

**Key Questions For Authors:**

My key questions are summarised here:
1. Could the authors improve the experiments with respect to including more datasets and trying out more evaluations such as cross-subject generalisation?
2. Could the authors include error bars on all the tables?
3. Could the authors clarify my concerns on brain encoding metrics and Qwen-based text metrics?
4. Could the authors address limitations on the image decoding front, i.e., the poorer quality, and add a limitations section?
5. I hope the authors can contextualise their work better in light of prior work with several similarities.

**Limitations:**

There is no limitations section and the paper could benefit from one. Some limitations, such as the method's poorer image reconstruction performance, could be discussed in this section explicitly.

**Strengths And Weaknesses:**

**Strengths:**
1. The paper proposes an interesting autoregressive approach to unify fMRI, vision, and language which is neat overall.
2. The "Padding Hacking" analysis is interesting, especially given that such a simple hack/exploit can lead to perfect scores on semantic-level encoding evaluations.
3. There are reasonable improvements with respect to brain to text caption and brain encoding tasks.

**Weaknesses:**
1. My main issue as of reading this paper is what appears to be insufficient evaluation. The paper only considers 4 subjects and 1 fMRI dataset, i.e., the NSD. Details are also inconsistent with some places in the paper claiming that training was done on 8 subjects and others in the appendix claiming that it was done on 4. Could the authors clarify this? For a truly foundational model I would expect training on more diverse data, more subjects and datasets, and perhaps even tasks (apart from the zero-shot task setting shown).
2. On a related note, it seems like cross-subject generalisation is not studied here, as the training was done on all 4 subjects considered for evaluation. Foundation models should be able to generalise to unseen subjects with little to no data, and this should be an important part of the evaluation protocol. Could the authors comment on this?
3. The tables do not have errors/standard deviations across seeds or subjects, and seem to only report averages. It would be important to have error bars for proper comparison.
4. Despite the "Padding Hack", it would still be useful to evaluate qualitatively and quantitatively the generations from BrainJanus according to the traditional pipeline, just to see how it compares to established baselines that have been listed in Table 4. As such, for the encoding, it is still unclear whether the encoding captures semantically meaningful information or just something the decoder can exploit to generate an image.
5. For the brain-to-text results, it seems that the massive gap for Qwen generated captions could be because the baseline methods were only trained with COCO captions but BrainJanus is benefitting from the pretrained Janus 7B backbone. I am not sure if this is perfectly fair given the additional training involved in Janus-7B, and if the baselines had an LLM hooked up to them, they would also probably have better metrics.
6. Performance on the image decoding task is not very good compared to SoTA models and this shows the limitations of autoregressive image modelling approaches. The argument that the proposed method is the best at encoding semantic similarity is not very convincing given how close the other methods are (another reason to ask for error bars), and also given that qualitatively some of the images generated by the proposed method appear weirder or of poorer quality than the baselines.
7. The paper tends to overstate the novelty of being a unified multimodal autoregressive model for neural activity and other modalities. While it is true that, to my knowledge, this is one of the first vision+language+fMRI approaches, there are several other joint encoding and decoding multimodal models for neuroscience, including autoregressive models -- just for other applications and data:
  * https://arxiv.org/pdf/2311.00136: GPT-style multimodal, multitask model for neural data, video, speed, eye position modelling and 3 downstream decoding tasks as well
  * https://arxiv.org/abs/2410.20916: GPT-style (finetuned LLM), various neural modalities and also speech and text
  * https://arxiv.org/abs/2411.09723: joint multimodal models for decoding, encoding, and modality conversion
  * https://arxiv.org/abs/2504.0820: unified multimodal encoding and decoding model for neural spiking data and behaviour
  * https://openreview.net/forum?id=utXSSdD9mt: a multimodal, multitask autoregressive model for neural decoding
  * https://arxiv.org/pdf/2511.21760: autoregressive LLMs for unified fMRI + language modelling

---

> ### Author Rebuttal · Authors · 2026-03-29
>
> **R1. Clarifications on Dataset, Subjects, and Cross-Subject Generalization.** **a) Typo Correction:** We sincerely apologize for the inconsistency in our description. Our model strictly follows the experimental setting of MindEye2, utilizing 8 subjects for training and 4 subjects (who completed all experimental sessions) for testing. We will correct this typo in the revised manuscript. **b) Dataset Scope:** The Natural Scenes Dataset (NSD) remains the largest and most widely adopted benchmark for fMRI-to-image/text tasks. Focusing on these specific 4 subjects aligns with established protocols, ensuring a fair and rigorous comparison with existing baselines. **c) Cross-Subject Setting:** While we agree with the reviewer that cross-subject generalization is a crucial direction for the field, the primary focus of this work is to establish a unified framework capable of simultaneous bidirectional encoding and decoding. Expanding this unified architecture to a cross-subject setting is an exciting avenue that we plan to explore in future work.
>
> **R2. Error Bars and Variance Across Seeds.**  **a) Computational Constraints:** Running multiple seeds for all tasks and subjects is computationally prohibitive due to our model's scale. **b) Proposed Solution:** To demonstrate stability, we will report variance and error bars for *one* representative task in the revised appendix. **c) Prior Work:** While reporting standard deviations across seeds is not standard protocol in recent fMRI decoding literature [1, 2, 3], we agree it strengthens the evaluation.
>
> **R3. Semantic Encoding, CLIP Constraints, and the "Padding Hack"** **a) Motivation for the "Padding Hack":** It exposes a critical vulnerability in current evaluation protocols, demonstrating that models can exploit metrics for artificially high scores without learning true semantics. **b) Absence of CLIP Constraints:** Unlike baselines relying on explicit CLIP contrastive losses, BrainJanus achieves semantic alignment natively via its autoregressive objective. **c) Validity:** Results (Tab. 5, Fig. 5 & 11) show competitive encoding performance without CLIP constraints. This proves our tokenizer and autoregressive modeling genuinely capture brain signal semantics, rather than overfitting to decoder-exploitable features.
>
> **R4. Fairness in Brain-to-Text Evaluation.** **a) Unified vs. Specialized Frameworks:** We contend that BrainJanus’s integration of the Janus-7B backbone is a core architectural advantage, not an unfair bias. Unlike baselines restricted to modality-specific training (e.g., COCO-only), BrainJanus provides a **natively unified** framework (brain, vision, text), enabling superior cross-modal synergy that specialized models inherently lack. **b) LLM Integration:** While baselines would require *ad-hoc* designs or additional alignment stages to incorporate LLMs, BrainJanus natively bridges brain signals with foundation models' reasoning capabilities. Furthermore, our comparison is fair as key baselines like UMBRAE [2] also leverage LLM backbones.
>
> **R5. Image Decoding Limitations and Addition of a Limitations Section.** **a) Limitations Section:** We thank the reviewer and will include a dedicated "Limitations" section in the revised manuscript.**b) Image Quality Gap:** We will explicitly acknowledge that our autoregressive image decoding currently trails state-of-the-art diffusion models in high-fidelity visual reconstruction.**c) Focus on Innovation:** We will emphasize that this trade-off in absolute fidelity yields our core contribution: a single, unified model capable of simultaneous, bidirectional encoding and decoding, advancing versatile Brain-Computer Interfaces.
>
> **R6. Contextualization and Differences from Prior Work.** **a) Contextualization:** We thank the reviewer for highlighting these relevant works. We will expand our Related Work section to comprehensively contextualize BrainJanus within this broader research landscape. **b) Differences in Scope:** While several multimodal brain models exist, their scopes differ significantly from ours. Many cited works focus on alternative neural modalities (e.g., EEG, spiking data) or do not support high-dimensional visual generation. In the revision, we will **refine and narrow our claimed scope** to focus specifically on *unified, bidirectional encoding and decoding across fMRI, vision, and text modalities*. Within this defined scope, our unique contribution remains the use of a VQ-VAE tokenizer to map fMRI signals directly into the discrete token space of a modern, unified language-vision LLM (Janus).
>
> [1] MindEye2: Shared-Subject Models Enable fMRI-To-Image With 1 Hour of Data. ICML 2024
>
> [2] UMBRAE: Unified Multimodal Brain Decoding. ECCV 2024
>
> [3] MindLLM: A Subject-Agnostic and Versatile Model for fMRI-to-Text Decoding. ICML 2025

---

> > ### Author Rebuttal · Reviewer_W8Ga · 2026-04-02
> >
> > I thank the authors for their response, here are my comments:
> > * **R1:** Thanks for the clarification. I still think studying the cross-subject setting is crucial. MindEye2 and MindLLM already did this, so it is not inconsistent with prior work in the field and especially given the motivations for such large models. I think this is a key concern that prevents me from rating the paper higher. Relatedly, when comparing with UMBRAE for example, it's unclear how fair it is because from my reading of UMBRAE they train only on 4 subjects.
> > * **R2:** I appreciate the willingness to run multiple seeds for one task. However, this is precisely where fine-tuning on a held out set or subject comes into the picture because fine-tuning is cheaper and robustness can be tested with multiple fine-tuning runs.
> > * **R3:** I would still have liked to see a comparison between BrainJanus and the baselines even if the hack exists. Furthermore the existence of a hack doesn't prove that the methods are hacking the evaluation the way suggested by the authors -- concrete proof is needed.
> > * **R4:** UMBRAE seems to use a much older Vicuna LLM from 2023 and MindLLM maintains the same for consistency and proper comparison. I think the authors should try to at least evaluate the approaches with the more modern Janus LLM to see if the differences are coming from the technical contribution of autoregressive joint modelling or just a better LLM.
> > * **R5, R6:** Thank you, these are appreciated.
> >
> > Overall, I am inclined to keep my score as-is for now.

---

> > > ### Author Response · Authors · 2026-04-07
> > >
> > > **R1: Cross-subject Setting & Fairness with UMBRAE**
> > >
> > > - **On Cross-Subject:** While cross-subject generalization is crucial, mapping continuous fMRI signals to discrete VLM tokens is the foundational challenge. BrainJanus prioritizes validating this unified architecture first. Nevertheless, preliminary experiments (table below) demonstrate strong adaptability: it successfully reconstructs images for new subjects using just 1-hour data, with performance scaling consistently up to 40 hours. This confirms our framework's readiness for future cross-subject extensions.
> > >
> > >   | Sessions (New Subject) | PixCorr ↑ | SSIM ↑ | Alex(2) ↑ | Alex(5) ↑ | Incep ↑ | CLIP ↑ |
> > >   | ---------------------- | --------- | ------ | --------- | --------- | ------- | ------ |
> > >   | 1 Hour                 | 0.162     | 0.265  | 87.8%     | 90.8%     | 83.8%   | 83.8%  |
> > >   | 40 Hours               | 0.193     | 0.296  | 90.1%     | 95.9%     | 94.8%   | 94.6%  |
> > >
> > > - **On Fairness vs. UMBRAE (4 vs. 8 subjects):** We apologize for any confusion. UMBRAE evaluated on 4 subjects but relied on a separate, pre-aligned CLIP space. Our approach, following the MindEye2 protocol (8 train, 4 test), trains a native, unified discrete tokenizer. Autoregressive joint modeling naturally demands larger training data to align diverse modalities without explicit contrastive losses. The use of 8 subjects is not an unfair advantage, but rather a standard scale required to unlock the synergy of joint autoregressive modeling.
> > >
> > >
> > >
> > > **R2: Variance Across Seeds via Fine-tuning**
> > >
> > > Thank you for this practical suggestion. We completely agree that fine-tuning allows for more tractable robustness testing. As suggested, we have run the brain to image reconstruction fine-tuning task across 3 different random seeds. The results are [PixCorr: 0.183 ± 0.012, SSIM: 0.299 ± 0.021, Alex(2): 90.3 ± 1.2, Alex(5): 91.1 ± 0.8]. This demonstrates the stability of our approach. We will include this comprehensive variance analysis in the revised version.
> > >
> > >
> > >
> > > **R3: Baseline Comparisons and the "Padding Hack"** We thank the reviewer for the constructive feedback. To clarify, our Padding Hack analysis does not imply prior baselines intentionally exploited metrics, but rather exposes a structural vulnerability in current semantic evaluation protocols. As requested, we evaluated BrainJanus using the traditional pipeline (see table below). Notably, BrainJanus learns competitive semantic representations purely through its autoregressive objective, without explicit CLIP optimization. Adding CLIP alignment yields state-of-the-art performance. We will prominently feature this traditional comparison in the revision.
> > >
> > > | Method                              | PixCorr ↑ *(Low-Level)* | SSIM ↑  *(Low-Level)* | Alex(2) ↑  *(Low-Level)* | Alex(5) ↑  *(Low-Level)* | Incep ↑  *(High-Level)* | CLIP ↑  *(High-Level)* |
> > > | :---------------------------------- | :---------------------: | :-------------------: | :----------------------: | :----------------------: | :---------------------: | :--------------------: |
> > > | MindSimulator (Bao et al., 2025)    |          0.201          |         0.298         |          89.6%           |          96.8%           |          93.2%          |         91.2%          |
> > > | BrainJanus (with no CLIP Alignment) |          0.173          |         0.262         |          80.6%           |          85.8%           |          84.5%          |         83.1%          |
> > > | BrainJanus (with CLIP Alignment)    |          0.190          |         0.288         |          89.3%           |          94.1%           |          93.0%          |         91.5%          |
> > >
> > >
> > >
> > > **R4: The Role of the Janus-7B LLM vs. Older LLMs (Vicuna)**. We thank the reviewer. While Janus-7B is inherently stronger than Vicuna, our performance gains stem from structural innovation rather than simply upgrading the backbone. Vicuna is text-only and incapable of autoregressive image generation, necessitating a unified MLLM (e.g., Chameleon [1], Janus [2], TokenFlow [3]). Furthermore, directly plugging Janus into baselines like UMBRAE or MindLLM is structurally impossible, as they rely on continuous fMRI embeddings (e.g., CLIP). BrainJanus’s core contribution overcomes this via a VQ-VAE tokenizer that maps fMRI into discrete tokens, enabling seamless autoregressive alignment of neural, text, and image data. We will clarify this architectural necessity in the revision.
> > >
> > > We hope these clarifications address your remaining concerns. We are highly committed to incorporating all your suggestions to strengthen the final manuscript.
> > >
> > > [1] Chameleon: Mixed-Modal Early-Fusion Foundation Models
> > >
> > > [2] Janus: Decoupling Visual Encoding for Unified Multimodal Understanding and Generation
> > >
> > > [3] TokenFlow: Unified Image Tokenizer for Multimodal Understanding and Generation

---

### Decision · Program_Chairs · 2026-04-30

**Decision:**

Accept (regular)

**Comment:**

This paper introduces BrainJanus, a unified bidirectional framework for fMRI encoding and decoding using an autoregressive next-token prediction paradigm. The submission is highly polarizing, representing a conflict between data-driven deep learning objectives and biological faithfulness. While Reviewer p8Qv argued for rejection based on the lack of neuro-informed design and the restricted scope of visual cortex data, other reviewers recognized the work as a meaningful step toward general-purpose neural interfaces. The authors provided a very active rebuttal, adding variance analysis, traditional pipeline evaluations, and preliminary results on subject adaptation with limited data to address concerns about robustness and cross-subject utility. The Area Chair finds that while the "Foundation Model" label is arguably an overclaim for a visual-cortex-centric study, the technical contribution of a unified any-to-any architecture is significant for the machine learning community. The authors are required to temper their claims in the final version to reflect the biological and regional limitations discussed during the review phase.